

# Generating synthetic fjord bathymetry for coastal Greenland

Christopher N. Williams[1], Stephen L. Cornford[1], Thomas M. Jordan[1], Julian A. Dowdeswell[2], Martin J. Siegert[3], Christopher D. Clark[4], Darrel A. Swift[4], Andrew Sole[4], Ian Fenty[5], and Jonathan L. Bamber[1]

[1]Bristol Glaciology Centre, School of Geographical Sciences, University of Bristol, Bristol, UK.
[2]Scott Polar Research Institute, University of Cambridge, Cambridge, UK
[3]The Grantham Institute, and Department of Earth Science and Engineering, Imperial College London, London, UK
[4]The Department of Geography, The University of Sheffield, Sheffield, UK
[5]Jet Propulsion Laboratory, California Institute of Technology, Pasadena, California, USA

*Correspondence to:* Christopher Williams (chris.neil.wills@gmail.com)

**Abstract.** Bed topography is a critical boundary for the numerical modelling of ice sheets and ice-ocean interactions. A persistent issue with existing topography products for the bed of the Greenland Ice Sheet and surrounding sea floor is the poor representation of coastal bathymetry, especially in regions of floating ice and near the grounding line. Sparse data coverage, and the resultant coarse resolution at the ice/ocean boundary, poses issues in our ability to model ice flow advance and retreat

from the present position. In addition, as fjord bathymetry is known to exert strong control on ocean circulation and ice-ocean forcing, the lack of bed data leads to an inability to model these processes adequately. Since the release of the last complete Greenland bed topography-bathymetry product, new observational bathymetry data have become available. These data can be used to constrain bathymetry, but many fjords remain completely unsampled and therefore poorly resolved. Here, as part of the development of the next generation of Greenland bed topography products, we present a new method for constraining the

bathymetry of fjord systems in regions where data coverage is sparse. For these cases, we generate synthetic fjord geometries using a method conditioned by surveys of terrestrial glacial valleys as well as existing sinuous feature interpolation schemes. Our approach enables the capture of the general bathymetry profile of a fjord in North West Greenland close to Cape York, when compared to observational data. We validate our synthetic approach by demonstrating reduced over-estimation of depths compared to past attempts to constrain fjord bathymetry. We also present an analysis of the spectral characteristics of fjord

centrelines using recently acquired bathymetric observations, demonstrating how a stochastic model of fjord bathymetry could be parameterised and used to create different realisations.

## 1 Introduction

Bed topography provides an essential boundary for modelling ice sheet dynamics, ice-ocean interactions and fjord circulation in Greenland (e.g. Vieli and Nick, 2011; Straneo et al., 2011). This widespread need for topographic information has motivated

the development of digital elevation models (DEMs) for the bed topography, which combine remote sensing measurements of the subglacial bed with the surrounding land and sea floor (Bamber and Layberry, 2001; Bamber et al., 2013; Morlighem et al., 2014). Each version of the Greenland 'bedmap' has provided improvements in resolution and reliability, with the most recent product to combine bed elevations and bathymetry data being Bamber et al. (2013) (from herein referred to as Bed2013). The



most recent Greenland-wide topography product (Morlighem et al., 2014) provides a significant improvement over previous versions towards the ice sheet margins. Despite these advances, and a substantial recent increase in the amount of observational data available (e.g. Jakobsson et al., 2012; Dowdeswell et al., 2014; Boghosian et al., 2015; Rignot et al., 2016), data coverage remains poor for many coastal regions. As a consequence, fjord bathymetry has not, in general, been well represented, and

non-physical discontinuities between land and ocean edges are apparent. In particular, in Bed2013 physically unrealistic morphologies arise at lateral boundaries of fjord mouths, as demonstrated by examples from the Greenland coast in Fig. 1.

To address these issues, the international research community has responded by collecting and compiling a wealth of new bathymetric data (e.g. Arndt et al., 2015; Boghosian et al., 2015; Rignot et al., 2016) with many other future campaigns planned

(e.g. the NASA Oceans Melting Greenland (OMG) mission). It will, however, take time for extended coverage to be achieved, and some fjord regions will likely never be surveyed due to both environmental and logistical limitations associated with operating in ice-infested waters. There is, nonetheless, an urgent need to better understand and model the processes that affect the dynamics of marine terminating glaciers in Greenland and elsewhere, thus requiring fjord bathymetry to be better constrained in DEMs.

Here, we present a new methodological framework for generating physically based fjord bathymetry in regions of sparse observational data availability. To provide context for the introduction of our method, we first present a review of existing geostatistical approaches to interpolating channel features in DEMs (including hydrological systems, palaeo-glacial troughs and subglacial channels). In particular, we describe why these methods are ill-suited to regions where sparse observational

data are available, which enables us to then demonstrate how our method provides a pragmatic solution to constraining the bathymetry of fjord systems. Our intent is that the presented approach will eventually be up-scaled to all unmapped fjords along the Greenland coast. This will significantly improve existing DEMs of bed geometry beneath and at the margins of the Greenland Ice Sheet as well as its surrounding surface topography and bathymetry. A novel feature of the method, which is inspired by analogue studies of glacial troughs (Coles, 2014), is the incorporation of predefined cross-sectional channel

geometry to provide a geometric structure that is physically realistic in the absence of observations, in turn providing realistic topography for applications including ice sheet modelling.

## 2   Past approaches for interpolation and integration of channel geometry in DEMs

For the purpose of integration in DEMs, fjords (Syvitski et al., 1987), river channels and glacial troughs (Batchelor and Dowdeswell, 2014) can be considered as pseudo-linear channel systems that have directional flow. In the absence of ade-

quate direct observations, the integration of anisotropic morphology is highly desirable when interpolating channel systems in DEMs. Where observations are available, there exist methods which can interpolate additional elevations of channel features (e.g. Herzfeld et al., 2011; Goff et al., 2014). However, where there are no data available, other than the known existence of a feature (discernible from remote sensing imagery), complications arise in how to accomodate the features in DEMs. In the





case of Greenland, the last data product to provide a continuous bed-to-bathymetry DEM (Bed2013) used different approaches to interpolate different topographic regions. Kriging interpolation was used for the interior of Bed2013. The bathymetry was taken from the International Chart of the Arctic Ocean (v3) (Jakobsson et al., 2012) referred to as IBCAO from this point forwards. The IBCAO DEM was developed from bathymetric observations using spline interpolation following Jakobsson et al. (2012). For Bed2013, triangulation (linear interpolation) was used to predict bathymetry within the fjords between the IBCAO and interior Greenland bed DEM datsets (Bamber et al., 2013), as these regions were unconstrained by observations. Using traditional isotropic interpolation approaches, such a lack of data often results in the generation of interpolated surfaces that fail to represent true channel geometry, and often appear artificially smooth. In the case of Bed2013, this problem resulted in the development of acknowledged physically unrealistic topographic artefacts (Fig. 1). For methods where anisotropy is not accounted for, and where observations are only available for small regions along a channel, interpolation can result in 'bulls-eye' anomalies (Dentith and Mudge, 2014), in which a channel is predicted as being a series of isolated basins (see Fig. 5(a) in Goff et al., 2014) as a result of clustering of the interpolation methods at observation locations.

To capture the appropriate geometry of channels, several different approaches have been developed involving geometric (e.g. Goff and Nordfjord, 2004; Merwade et al., 2005), mathematical (e.g. Herzfeld et al., 2011) and mass conservation (Morlighem et al., 2014) solutions. To place our study in the context of other interpolation methods we review previous approaches with a particular focus on resolving curvilinear features (channels). Additionally, stochastic perturbations to Greenland Bed DEMs can be employed in a variety of different ice-sheet modelling contexts (cf. Durand et al., 2011; Seroussi et al., 2011; Sun et al., 2014). It is possible that there will be a future need for similar stochastic modeling of fjord bathymetry, and we also discuss this here.

## 2.1 Kriging

The key issue with interpolating features for which orientation is important (e.g. channels) is the ability to incorporate direction into the method used to develop them from observations. Kriging — a method of interpolation for which the interpolated values are modeled by a Gaussian process — is often employed to create continuous surfaces from point data (e.g. Hock and Jensen, 1999; Bamber and Layberry, 2001; Le Brocq et al., 2010; Bamber et al., 2013). The approach accounts for the statistical properties of observations within a local search neighbourhood using a variogram function (Deutsch and Journel, 1998) and, using this it is possible to incorporate various types of anisotropy within the basic framework (Merwade et al., 2006). However, the method only holds when applied over regions sharing the same overall statistical properties whether that be, for example, the same geologic rock type or the same directional bias. When anisotropy is defined relative to a fixed Cartesian coordinate system, and where data are sparse, kriging is impractical for sinuous features with constantly varying direction such as channels (see also Fadlelmula F. et al., 2016). Specifically, dividing a region into areas of shared anisotropy (thus satisfying the assumption of stationarity within a search window) that are data sparse prevents the adequate population of the variogram with which to statistically model the region.



## 2.2 Channel coordinate transformations

To enable interpolation across channel widths, one approach uses cross-sectional profiles, but to do so, typical channel sinuousities present a problem. As an intermediary step to interpolating sinuous channels in DEMs, several approaches have been developed to transform the coordinate system of a given channel — moving from Cartesian coordinate space to channel

coordinate space — enabling removal of complex sinuosity and the creation of an artificially straight channel (cf. Goff and Nordfjord, 2004; Merwade et al., 2005). Channel space (sometimes denoted as $s, n$ in the literature) differs from Cartesian space in that locations are transformed relative to their distance along the channel ($s$) and perpendicular to the centreline ($n$). Observations within channel space — a now straightened channel — can be locally interpolated by considering a single direction as opposed to a continuously changing one. The interpolated channel can then be transformed back to Cartesian space.

The approach breaks down, however, where multiple channels merge together at confluences. Furthermore, in the absence of sufficient observations, such an approach cannot be used alone to predict along-channel geometry without additional interpolation. For example, manual digitisation has been applied to individual channels to assist in the development of a realistic bed topography for Thwaites Glacier, West Antarctica (Goff et al., 2014). Additionally, channel straightening through coordinate transformation becomes difficult where channels manifest high levels of sinuosity or sharp changes in direction (Goff and

Nordfjord, 2004).

## 2.3 Mathematical morphology

Further issues with regard to maintaining morphological characteristics of channels, in particular ensuring known depths are honoured, are apparent in large and low resolution datasets particularly where interpolation methods are applied (Herzfeld et al., 2011). Where resultant data products are to be used in modelling studies, honouring known maximum depths is key as incorrect

values can adversely affect results - especially with regard to maximum and minimum elevations (Herzfeld et al., 2011). To ensure true morphology is maintained, Herzfeld et al. (2011) proposed a routine which initially interpolates glacial channels along a mean direction vector. Connectivity between points along the trough is then established and the locations of gridded points are adjusted to be within the vicinity of the now-defined channel. Elevations are then mapped with minimum elevations being applied to adjusted points now in the channel. This 'mathematical morphological' approach is effective in regions where

observations (gridded or not) covering features of interest are available. The adjustment of gridded points to follow channel directions provides a succinct approach to avoid the constraints of regular gridding, which mask channel structures especially at lower resolutions. However, observations are required to identify channels and application of the 'mathematical morphology' approach becomes complicated in the case of multiple interconnected dendritic type networks.

## 2.4 Mass conservation

Subglacial channels, which occur beneath grounded ice, are significantly easier to interpolate into DEMs than fjords as a physically based mass conservation optimisation scheme can be applied (Morlighem et al., 2011, 2014). This approach is independent of traditional geostatistical interpolation methods. Bed elevation values are calculated from ice thickness values,





which are derived from combination of radar sounding measurements and surface velocity observations and of course using the assumption that mass is conserved along flow. Despite such an approach being useful for subglacial channels covered by grounded ice, this approach cannot be applied for regions of open ocean or non-grounded ice as is the case for fjords and cross-shelf troughs on formerly glaciated continental shelves.

## 2.5 Remaining issues

Despite the approaches that have been developed to interpolate channels in DEMs, there are a number of recurring problems in applying these methods in the next generation of the Greenland DEM. In particular, all of the methods assume that there are at least some data from which to extend and predict the structure of a given feature. Furthermore, no method is explicitly designed to include or represent the known physical characteristics, in particular the cross-sectional geometry, of the particular type of channel system (e.g. U-shape of glacial; v-shape of fluvial), with morphological information only being extended from available observations. Thus, there remains a disconnect between the presented frameworks and cases where features (1) are known to exist; (2) are assumed to conform to a structure related to the processes by which they were created (e.g. an assumed u-shape in the case of fjords where no other data are available); and (3) have no observations available to provide geometric constraints. A framework for fjord channel systems which addresses these issues, and can be applied to a large area such as the Greenland coast, must be able to:

- Impose morphological geometry to features of known process-origin;

- Account for elevation trends along and across the channel;

- Account for confluences in dendritic channel systems;

- Enable repeatable application across numerous channels within dendritic systems;

- Be able to deal with minimal data input (other than absolute limits e.g. minimum and maximum depths as well as spatial extent).

## 2.6 Stochastic models

Stochastic models of bathymetry have long been employed to abyssal hill features in the deep ocean (e.g. Goff and Jordan, 1988). In such places, stochastic models are appropriate for use because the frequency power spectra of deep ocean bathymetry follows well defined parametric relationships (Bell, 1975). Specifically, the high-frequency tail of the power spectra is characterised by power-law relationships (i.e. the Brownian regime, which can be stochastically modelled), with lower-frequency behaviour characterised by a flat spectra (i.e. the white regime, which cannot) (Goff and Jordan, 1988). This spectral behaviour is common across other types of natural terrain and, subsequently, spectral analysis of natural terrain often focuses upon establishing the transition between high- and low-frequency behaviour, and the characterisation of the high-frequency power-law relationships (Shepard et al., 2001). To the best of our knowledge, there are no studies that have considered the spectral analysis





of fjord bathymetry. As part of this study, we use data that are available from surveyed fjords to constrain the stochastic models of the bathymetry of many Greenland fjords.

## 3    Methods

A flow diagram for the separate components of our method for generating physically based fjord bathymetry is presented in
Fig. 2. Each component is described in a separate sub-section. In Sect. 3.1 we discuss the approach taken to map the centreline of each fjord within the fjord system introduced below. Using the mapped centreline, we explain in Sect. 3.2 how a point mesh is developed, populating a given fjord with points extending from the centreline to the fjord edges based on the Greenland Ice Mapping Project (GIMP) land classification mask developed from remote sensing imagery (Howat et al., 2014; Morlighem et al., 2014). Elevations are then associated with the points within the mesh, incorporating an assumed parabolic cross-profile geometry, described in Sect. 3.3. The elevation dataset now developed is then used to create a continuous surface, representing the fjord bathymetry in Sect. 3.4. Finally in Sect. 3.5 we describe a stochastic modelling approach based on recently acquired observational data (Rignot et al. 2016, OMG Mission 2016) the data being referred to as OBS1516 from this point forwards. The synthetic realisations within this study are based on two datasets - IBCAO and OBS1516. Consequently, we differentiate these simulations by naming them SynthIBCAO and SynthOBS respectively.

The sequential approach defined in Fig. 2 was applied to a fjord system in North West Greenland close to Cape York (see Fig. 3) for which we identified and mapped the centrelines of five individual fjords. This fjord system was recently surveyed (OBS1516) from which a DEM is now available and allows for a comparison between our synthetic generation method and *insitu*, high resolution (150 m) gridded observations.

## 3.1    Centreline mapping

The ability to map a given fjord where no observations are available requires the provision of a skeleton mesh, which hinges on the presence of a centreline - an imaginary line that is equidistant from the two fjord edges. Consequently, the first step in the synthesis of a given fjord's geometry requires a centreline to be defined. Approaches exist for automatic centreline identification for glacier surfaces (e.g. Kienholz et al., 2014; James and Carrivick, 2016) as a means of avoiding manual digitisation. Such applications are, however, informed by the availability of a glacier surface elevation DEM. An equivalent non-geomorphologically based method includes the definition of the medial axis (cf. Blum, 1967) or topological skeleton and is frequently used in image processing and computer graphics applications (see Bai et al., 2007). Various packages are available to calculate topological skeletons (e.g. Van Der Walt et al., 2014). However, these algorithms are based purely on an input image and are sensitive to image pixel resolution. For our intended application, this can result in the development of a centreline (or skeleton) with multiple branches along a single channel feature.





The centreline mapping method that we introduce allows fjord systems with multiple branches to be accounted for. Each centreline extends from a predefined seed point (or points) at the head of the fjord, ending at a predefined end zone (e.g. the fjord mouth). The centreline itself is defined by a series of points or vertices, each with a unique identifier. Fjord confluences and the implementation of network structure are described in Sect. 3.4. We define the centreline as being any path between a seed and the end zone which minimises the path integral whilst maximising the distance of the path from the fjord walls. This removes the issues of multiple side branches that arise using existing skeletonisation approaches. The algorithm incorporates direction and thus an aspect of evolutionary landscape process knowledge, which ensures that the centreline captures a leading order feature from the landscape it represents. Furthermore, this approach ensures that the paths and vertices are given unique identifiers enabling them to be specifically referenced, which is important when defining the channel mesh (see Sect. 3.2).

Fjords were identified as channels between areas of land and ice leading towards the open ocean, identified here using the modified GIMP land classification product (Howat et al., 2014; Morlighem et al., 2014) (see Fig. 3(a)). At the head of each fjord, multiple seeds were manually created from which to initiate a path. The end target was defined as a broad region rather than a specific point (in this case, the edge of the land classification mask). The following algorithmic steps were then undertaken:

1. Using the land classification mask, we calculate the distance transform between land/ice and ocean ($d$), from which the shortest distance of any location within the ocean from regions of land or ice can be identified (see Fig. 4).

2. Based on the slope of the distance-transform calculated for regions of ocean relative to land/ice land categories using GIMP (Fig. 5(a)), and considering the edges of the fjord, the initial seed points generate new points at a predefined distance interval which propagate along the fjord (see Fig. 5(b)). Up to four new nodes are generated at each step, such that the angle between the newly defined edge and the parent edge is less than $\pi/6$, the angle between any pair of new edges is no less than $\pi/24$ and the new edge does not cross the fjord boundary. If we knew that no path would branch, we could generate a single new node: this more complicated procedure is adopted because we do need to consider branches. If more than four nodes are generated in this manor, then those with the smallest values of $L = 1/d^4$ (distance from the fjord edges) are selected.

3. Where, for example, a seed generates three new points, this results in the creation of three paths. Paths then increase in length as more points are created, with new paths following the creation of each new point. In the example illustrated in Fig. 5(b), the initial seed creates three new points, each along a separate branch: 1.1, 2.1 and 3.1, each of which spawned its own new points and resultant branches i.e. 1.1.1, 1.1.2 etc.

4. The process in step 3 alone would lead to exponential growth in the number of paths. To avoid this, paths are culled frequently (every three generations). Each path is categorized into bins ($x_i, y_j, a_k$), where the centroid of the path $x, y$ satisfies $|x - x_i|, |y - y_j| < 16$ km, and the angle defined by the last edge added to the path, $a$, satisfies $|a - a_k| < \pi/8$. The path with the lowest value for the path integral of $L$ is retained from each bin, and the remainder discarded.





5. Where a path meets a boundary that is not the predefined end zone (e.g. land), the path is culled, as illustrated for branches 2.1.1 and 2.1.2 in Fig. 5(b) within the pink box. In this example, as a consequence of the boundary intercept, there is a resultant culling of both paths 2.1.1 and 2.1.2, following the removal of the parent node 2.1.

6. Once a given path reaches the target region, its length is compared to the length of all other complete paths with only the shortest being retained. In Fig. 5(b), paths 1.11.1, 1.11.2 and 1.11.3 complete, the shortest where $L$ is minimised (1.11.2) being retained and used to define the fjord centreline.

7. A centre of mass (COM) is calculated for each path. When considering the length of all complete paths, where the distance between the COM of separate paths is greater than a threshold value (manually set to $\sim$ half of the mean channel width in an area), both paths are kept regardless of length, otherwise the shorter, where $L$ is minimised, of the two paths is retained. The use of the COM allows for separate centrelines to be defined along more complex fjord networks than by culling according to path length alone.

## 3.2 Fjord mesh development

For each fjord centreline, points normal to each centreline vertex were defined up to the fjord edge taken from the GIMP land mask (Fig. 6). The angle of the normal vector along which these new points were defined was calculated from its orthogonal relationship with the vector joining the neighbours of a given centreline vertex. To avoid an irregular distribution of new points in the interpolated profile at the channel edges, the points used to define the vector from which the normal was calculated were sometimes selected from more distant neighbours. This was particularly pertinent at more sinuous sections of a centreline (see Fig. 6). This smoothing of the profile is adapted from Goff and Nordfjord (2004). Vertices normal to the centreline were calculated up to the mouth of the channel, at which point the fjord centreline was manually clipped.

## 3.3 Mesh elevation definition and cross-sectional fjord geometry

Elevations were attributed to the point mesh by first constraining the seed and fjord-edge bed elevations through association with the nearest bedrock/bathymetric observation (either from ice penetrating radar where ice covered (see Bamber et al., 2013); altimetry where exposed bedrock (Howat et al., 2014); or from bathymetric observations (OBS1516)). This method ensured that at the head of the fjord, three elevations were available — the two edges of the fjords (taken as the elevations at the first land locations encountered at the fjord edges) and a centreline elevation. For future applications along the Greenland coast where seed data are sparse, modelled estimates from the mass conservation optimisation scheme (Morlighem et al., 2014) could be used.

Two approaches were taken to assign bed elevation values along the centreline of a given fjord. In both approaches, bed elevations were linearly interpolated between a known bed elevation at the head of the fjord (taken from OBS1516 (Fig. 3)) and a known bed elevation at the mouth of the fjord - the mouth having been manually located and consistently used for all model runs. For the first run (SynthIBCAO), the bed elevation at the mouth was taken from the nearest IBCAO observation (20





km from the mouth of the fjord system depicted in Fig. 3) and was set at -803 m. This was chosen for the first simulation as until recently IBCAO provided the most extensive bathymetric dataset for Greenland and the distance from a fjord head to the nearest observation is often $\sim$ 10s of kilometres. For the second run (SynthOBS), the gridded bathymetric observation from OBS1516 at the same position was used (-920 m — see Fig. 3). Should high frequency stochastic perturbations wish to be

added along the profile (see Sect. 3.5 and 4.5), they would be applied at this stage. Bed elevations up to the termini of most glaciers in Greenland, albeit predominantly modelled, are now available (Morlighem et al., 2014). We justify the use of the OBS1516 data for defining the elevations at the head of each fjord in the presented simulations as it enables a comparison of synthetic and observational data directly, removing the need to consider uncertainties inherent of modelled elevations.

In the absence of large scale studies on fjord bathymetric geometry, we base our cross-sectional fjord geometry on the prior analysis of over 8000 glacially eroded valleys now exposed by interglacial ice-sheet retreat (Coles, 2014). In their study, profiles were acquired from different glacial and geological environments including valleys from the Southern Alps (New Zealand), the Pyrenees and North and South Patagonia. For the valleys the bed elevation, $V_d$, was fitted to a power law relationship for the form

$$V_d = \alpha |w|^\beta, \tag{1}$$

where $\alpha$ is the form ratio (valley depth/valley top width), $w$ is the distance along the cross-section from the centreline (the position of which corresponds to $w$=0), and $\beta$ is the power law exponent. Best fit parameters, of $\alpha$=0.20, $\beta = 1.38$ were obtained (Coles, 2014). A value of $\beta = 2$ (i.e. a parabolic relation) follows Wheeler (1984).

Equation 1 assumes that a given fjord's cross-section is symmetrical about the centreline with the centreline as the deepest point; an assumption which usually does not hold exactly. Additionally, the fjords are often seeded with edge elevation data that are significantly higher on one side of the fjord than the other. To define the cross-sectional fjord geometry, we define a parabola of the form $ax^2 + bx + c$ (Wheeler, 1984), where $a$, $b$ and $c$ are calculated based on the known elevations and relative locations of the edge and centreline points. This enables us to relax the constraint that the centreline must be the deepest point.

Thus, the parabola used to define across-fjord geometry in this study is inspired by the analyses of Coles (2014) but not a direct application of it.

### 3.4 Fjord surface generation, implementation of fjord confluences, and wider integration with DEMs

Following the development of a complete fjord elevation mesh, a surface was then made for each fjord, with mesh point elevations being mapped to a regular grid, thus creating a continuous surface. The resultant grid was then masked using the

GIMP mask, thus removing any values outside the extent of the point mesh which arise as a result of the regular gridding process. The individual fjord grids were then combined, from which a final grid of the minimum values (or maximum depths)




was created. Thus, the lowest value at any location where two fjords overlap was retained (Goff and Nordfjord, 2004). As a result, deeper grid values took precedence over those that were shallower. This approach coupled with the aforementioned setting of edge elevations at confluence locations avoids the creation of ridge artefacts in the final DEM. The fjord DEM was then integrated into the wider landscape DEM (Bed2013) which includes non-fjord regions. Prior to the merge, Bed2013 was

masked, removing any values in the area occupied by the synthetic fjord(s).

### 3.5 Stochastic modelling of fjord bathymetry

In this section we describe spectral analyses methods used to constrain the fjord's statistical features, and the inverse methods that can be used to generate synthetic profile. Our analysis is based upon analogous analysis of abyssal hill features in the mid-ocean (Bell, 1975; Goff and Jordan, 1988), although it is simpler in the respect that fjord bathymetry is approximated as

a one dimensional problem. Using the centreline mapping approach presented in Sect. 3.1, centreline points were established, with vertices on a 150 m interval for nine fjords along the west Greenland coast, selected where mean gridded observations were contiguous along fjord centrelines according to OBS1516 (Fig. 7).

The lengths of the nine fjord sections in our example are constrained by the length of the shortest fjord section (~30 km).

Elevations were extracted for each centreline vertex, providing one dimensional centreline elevation profiles. Where a centreline contained missing data at a level no greater than 20 %, a cubic interpolation routine was applied to give a continuous elevation profile (Fig. 8). Prior to performing the spectral analysis, each elevation profile was linearly detrended, which acts to emphasise the overall variation of the small scale trends (Shepard et al., 2001). Each elevation profile was then transformed using the numerical Fast Fourier Transform algorithm converting it to the frequency domain (Van Der Walt et al., 2011). Power

spectra for each fjord were then obtained from the square of the complex modulus, and arithmetically averaged over the nine selected fjord profiles, to create a composite power spectra. This arithmetic averaging approach is as described in Bell (1975) for mid-ocean bathymetry, and enables longer wavelength features to be statistically constrained along with the higher frequency features that are repeatedly sampled.

Of interest in this study is demonstrating, in a proof-of-concept manner, how the composite power spectra for the fjords can be used to generate different one dimensional realisations of synthetic bathymetry, that are consistent with the overall statistical properties. In order to generate the different realisations of bathymetry, we use the inverse Fourier transform method outlined by Tyan et al. (2009) (the sinusoidal approximation method described in Sect. 4 of their study). Their method was introduced in the context of generating one dimensional random road profiles, which is a mathematical analogue of fjord

profiles. In their formulation the Fourier amplitudes of each harmonic are determined by the power spectra of the profile, with stochasticity present via the random relative phase of each harmonic. Our only modification to their method is to use a different parametric form for the power spectra, which is motivated by our observed results described in Sect. 4.5 and is consistent with the generality of their method.





## 4 Results

In this section, we present differences between Bed2013 (the last Greenland bed-bathymetry combined data product following Bamber et al. (2013)), OBS1516 (recently acquired fjord bathymetry data), and synthetic fjord bathymetry developed using the methods described in Sect. 3.1 — 3.4. The first synthetic application is preconditioned using the nearest IBCAO bathymetric

observations (SynthIBCAO) and the latter using the OBS1516 dataset (SynthOBS). The results are compared to Bed2013 and OBS1516, in the region illustrated in Fig. 3. The extent of OBS1516 relative to the area of interest is displayed in Fig. 9(a). Spectral analysis of selected fjord profiles shown in Fig. 8 is then presented, following application of the method described in Sect. 3.5 to available fjord bathymetry data in the region illustrated in Fig. 7.

### 4.1 Bed elevation differences for OBS1516 vs. Bed2013

Prior to investigating the improvements of the synthetic algorithm over the existing Bed2013 DEM within the area of interest proximal to Cape York (Fig. 9(a)), we consider areas of maximum over- and underestimation of bed-elevation that are present in Bed2013 within the region covered by OBS1516. Bed2013 is a continuous DEM extending from the bed beneath the contemporary ice sheet out to the continental shelf in the ocean, with all bathymetric information derived from IBCAO.

IBCAO was combined with the bed elevation component of Bed2013, with triangulation used as an interpolator to provide values where IBCAO was unconstrained by data (Bamber et al., 2013). Triangulated portions of the resultant DEM were then smoothed using a 2 km window (Bamber et al., 2013). Where there was an unrealistic offset between the two surface datasets (e.g. bathymetry was higher than the glacier bed), some areas were manually dropped to force them to adhere to a subjectively more realistic profile (i.e. a fjord would be lower than the glacier bed upstream of it). The result of differencing Bed2013

from the OBS1516 data set is presented in Fig. 10(a) with the frequency distribution of the differences presented in Fig. 10(b). On average, Bed2013 underestimated the depth of OBS1516 by 88 m, for which a skewness of -0.96 from the difference frequency density distribution was identified. These overall dataset statistics obscure the regions of maximum depth underestimation which are focused within the fjords themselves. Absolute maximum under- and overestimates of OBS1516 by Bed2013 reached 934 m and -1478 m respectively. Regions containing these extreme values can be directly associated with portions of

the IBCAO dataset that were unconstrained by observations and were themselves the result of triangulation (Bamber et al., 2013) and spline interpolation (Jakobsson et al., 2012).

### 4.2 Bathymetry for SynthIBCAO

The first implementation of the synthetic fjord routine, SynthIBCAO, defines the elevation at the mouth of the fjord based on

the nearest IBCAO bathymetric observation, with the elevation of the point at the head of the fjord taken from the OBS1516 dataset . Points normal to each centreline vertex were then calculated as described in Sect. 3.2 and 3.3. The resultant combined





surface DEM with the inclusion of synthetically created fjord bathymetry is displayed in Fig. 9(c).

The SynthIBCAO channel geometry is both deeper and more concave compared to Bed2013 (Fig. 9(b)), particularly with regard to the narrower fjord regions. Based on the contour pattern (Fig. 9(c)), these narrower fjord regions now display a deeper

and more concave cross-sectional profile than was rendered in Bed2013. For the wider confluence region centred south from (-705, -1340) on Fig. 9(c), there is a clear change in the overall depth profile with SynthIBCAO reaching -731 m compared to -391 m in Bed2013. SynthIBCAO reaches a minimum bed elevation difference to the defined elevation at the mouth of the fjord (-803 m) as a result of the regular gridding of the fjord mesh elevations described in Sect. 3.4.

Comparing the difference between Bed2013 and SynthIBCAO (Fig. 11(a)), the latter dataset has elevations consistently lower than the former. The mean offset between the two datasets was 274 m. The changes along the narrower portions of the fjords — up to $\sim$ 3 km from each respective fjord head (see Fig. 3(b) for mapped channel centrelines) — are relatively small ($\sim$ 0 – 50 m). Larger offsets are apparent where fjords enter the confluence region centred south from (-705, -1340) on Fig. 11(a), with a maximum offset of 547 m. The increased concavity of SynthIBCAO is well illustrated with a mean increase in

depth along the confluence zone centreline of $\sim$ 370 m.

Subtracting SynthIBCAO from OBS1516 (Fig. 11(b)) reveals a mean offset between the two datasets of around 50 m. Relatively good agreement along the first $\sim$ 3 km of each fjord (see Fig. 3(b) for mapped channel centrelines) is displayed, and indeed portions of the main confluence region, with differences centred at 0 m. The main region of elevation overestimation

(i.e. lower than the observations) is focused at the confluence point of the fjord 1 (refer back to Fig. 3(b) for fjord numbers), the region of overestimation focused around (-709, -1344) on Fig. 11(b) up to a value of $\sim$ 580 m. This overestimate is likely indicative of the presence of a sill-like feature. The two main regions of depth underestimation using SynthIBCAO are centred at (-704, -1338) and (-704, -1355) on Fig. 11(b), with maximum underestimates of $\sim$ 385 m and $\sim$ 358 m respectively. These underestimates possibly relate to the presence of overdeepening type features present in OBS1516.

For reference, a comparison with OBS1516 subtracted from Bed2013 for the same area of interest is drawn (Fig. 11(e)). As described in Sect. 4.1, Bed2013 consistently underestimates bed elevation. However, as with SynthIBCAO, the main areas of underestimation are focused at the same locations - namely (704, -1338) and (-704, -1355) on Fig. 11(e) for which overdeepenings are likely present.

## 4.3  Bathymetry for SynthOBS

The second implementation of the synthetic fjord routine, SynthOBS, defines the elevation of points at both the head and mouth of the fjord based on gridded elevations from OBS1516 at the same location. The resultant combined surface DEM with the inclusion of synthetically created fjord bathymetry is displayed in Fig. 9(d). SynthOBS demonstrates deeper concave geometry across the fjords compared to Bed2013. The changing relief of the banks of the synthetic fjords are steeper than those rendered





in the original DEM (Fig. 9(b)). Between fjords, there are also changes in the elevations of the ridges such as at (-706, -1330) on Fig. 9(d). The differences between the synthetic and the original DEMs are further quantified by the difference plot illustrated in Fig. 11(c).

The SynthOBS surface is generally lower than Bed2013 (Fig. 11(c)), with a mean offset between the two datasets of 316 m. The only locations where SynthOBS was higher than Bed2013 were at the edges of the fjords within ∼ 3 km from each respective fjord head (see Fig. 3(b) for mapped channel centrelines). This possibly highlights overly smoothed sections of Bed2013 (where it was combined with the IBCAO dataset - see Bamber et al. (2013)). As with the SynthIBCAO approach, the largest offsets are apparent as fjords enter the confluence region centred south from (-705, -1340) on Fig. 11(c), with a mean
offset in this region of ∼ 400 m.

    Subtracting SynthOBS from OBS1516 (Fig. 11(d)), the mean offset between the two datasets is ∼ -3 m. The spatial pattern is very similar to that described for SynthIBCAO with the same regions of under- and over-estimation being equally apparent. Specific values, however, differ. Maximum overestimation focused around (-709, -1344) on Fig. 11(d) was ∼ 581 m. The two
main regions of elevation overestimation (i.e. higher than the observations) using SynthOBS centred at (-704, -1338) and (-704, -1355) on Fig. 11(d) have maximum underestimates of ∼ 328 m and ∼ 283 m respectively. Overall, the profile represented by SynthOBS is closer to OBS1516, as would be expected considering the elevation profile of each fjord was calculated between fjord head and mouth observations from the OBS1516 dataset.

### 4.4    Centreline profile changes: Bed2013, OBS1516, SynthIBCAO and SynthOBS

Considering centreline profiles for all fjords, the improved general elevation profile of each fjord using the synthetic approaches — SynthIBCAO and SynthOBS — relative to OBS1516, as well as the underestimation of elevation in Bed2013, are illustrated in Fig. 12.

    The synthetic realisations underestimate observed bathymetric elevation to a much lesser extent than Bed2013, capturing
the generally sloping profile of OBS1516. The good agreement (approximately ± 50 m) of synthetic–observed values along the first ∼ 3 km of each fjord – in particular for fjords 1,2 and 5 – implies the presence of approximately linear profiles. Larger differences — indicative of where the synthetic approach performs less well — occur from ∼ 4 km along each centreline, which relates to the confluence region of the individual fjords (refer to Fig. 3). Higher frequency features (along track peaks and troughs likely relating to sills and overdeepenings) are not captured using the presented synthetic fjord bathymetry generation
approaches.

### 4.5    Spectral characteristics of observed fjords

Following Sect. 3.5, we now consider the spectral characteristics of the fjord bathymetry along the centrelines of the nine fjords illustrated in Fig. 7, using the OBS1516 data. A log-log plot for the mean power spectra, $S(k)$ where $k$ is the wave





number (linear spatial frequency), is illustrated in Fig. 13 (blue crosses). The power spectra exhibits an approximate power-law relationship at higher frequencies (corresponding to a linear relationship in log-log space), and an approximate flattening at lower frequencies. A parametric model which captures this frequency transition is:

$$S(k) = \frac{F_0}{k^\alpha + k_0^\alpha}, \tag{2}$$

where $k_0$ represents the approximate transition frequency between the high and low frequency regimes, $\alpha$ is the exponent for the high frequency tail (for $k >> k_0$, $S(k) \propto k^{-\alpha}$), and $F_0$ acts as a normalisation constant. The parametric model (equation 2) is a generalisation of the model for the power spectra of ocean bathymetry in Bell (1975), which assumes $\alpha$=2. In general, different types of natural terrain can exhibit a range of spectral exponents (Goff and Jordan, 1988; Shepard et al., 1995, 2001), and our parametric model is representative of this.

The parametric best fit values were obtained using a non-linear least squares solver, and correspond to $F_0$=17.6 m $^2$ km$^{-1}$, $k_0 = 0.069$ km$^{-1}$ and $\alpha$=1.74 (Fig. 13, red solid line). The transition frequency, $k_0 = 0.069$ km$^{-1}$, corresponds to a transition wavelength of 14.5 km. This compares with a transition spatial frequency $k_0$=0.025 km$^{-1}$, and a transition wavelength 40 km, for abyssal hill features in the mid ocean in Bell (1975).

Figure 14 shows two different realisations of synthetic fjord bathymetry using the parametric fit to the power spectra in Fig. 14, and the stochastic inverse Fourier transform procedure describe in Sect. 3.5. The horizontal spacing of the synthetically generated profiles is set to be the same as the bathymetric data (0.2 km). If we draw a comparison between the stochastic model of synthetic fjord centreline profiles and the OBS1516 profiles (Fig. 12), it is clear that the synthetic profiles do not contain
the lowest-frequency oscillations (wavelengths $\sim$ 15 km or greater). This is consistent with the general flattening of the fjord power spectra at low frequencies. However, oscillations on a length scale $\sim$ 5 km (typical of sills and overdeepenings) are present in the synthetic profiles, albeit the specific locations of such features in these profiles are random.

## 5   Discussion

Despite the concept of channel elevation point meshes not being new in hydrology (Merwade et al., 2005, 2008), and in some glacial studies (Goff et al., 2014), this study is the first time such an approach has been applied to large fjord systems. A key addition presented in this study, which addresses sparse data availability, is the introduction of parabolic cross-sectional form along each profile that is characteristic of glacial fjords. In the absence of data, continuous DEM surfaces are developed using interpolation procedures. The specific values assigned to regions lacking observations are thus entirely dependent on the
interpolation routine applied and the presented approach provides a physically based estimate of elevations in these regions.





The introduction of the artificial mesh removes the need to apply a traditional interpolation routine over a large region, instead providing an idealised mesh to constrain regions known to be fjords. The method presented must, however, be semi-informed by data. The minimum elevations that are required are the fjord bank edges (i.e. topographic elevation at the land/ocean inter­face according to a land mask e.g. GIMP) — which in general can be different from one another — as well as the elevation

of the assumed centreline. The deepest point along the channel is constrained by the quadratic fit. In the case of Greenland, for which this method has been developed, ice-free edge observations are widely available (e.g. Howat et al., 2014; Korsgaard et al., 2016). Equally, observations at the head of the fjord can be taken from bed elevations inferred from mass conservation (Morlighem et al., 2014), or, in some regions, radar observations (e.g. Gogineni et al., 2001). Finally, observations for the fjord mouth could be taken from datasets including IBCAO or others (e.g. Schjøth et al., 2012; Dowdeswell et al., 2014; Arndt et al.,

2015; Rignot et al., 2016), albeit that these values may be a significant distance from the fjord mouth itself, which using the presented approach may result in further under- or over-estimation of a given fjord centreline elevation profile.

The synthetic approaches — SynthIBCAO and SynthOBS as presented in Sect. 4.2 and 4.3 respectively — represent two sit­uations that would be encountered when applying the method, as part of wider Greenland DEM development, to fjords around

Greenland. By informing the mouth elevation on IBCAO observational data at a distance of ∼ 20 km from the mouth, the im­pact of using distant bathymetric observation is exemplified. Equally, as many fjords have at least some information following various recent campaigns (including Schjøth et al., 2012; Dowdeswell et al., 2014; Arndt et al., 2015; Rignot et al., 2016), the use of observational data to constrain the algorithm is illustrated by SynthOBS. The application of these two synthetic approaches has provided bathymetry more representitive of the observed elevation profiles (OBS1516) of fjords within the

area of interest (Fig. 12). Within this region, topographic features, such as sills and overdeepenings, captured within OBS1516 occur. It is not possible to predict oscillatory features such as these with the geometrically flat surfaces assumed by our basic algorithm. In these examples, the overdeeping features and sills have a length scale ∼ 5 km which is less than the transition wavelength for the fjord power spectrum of ∼ 15 km (Fig. 13). The transition wavelength provides an approximate upper bound upon the length scale of features which could be modelled using our stochastic framework. Subsequently if we integrated the

analysis here with the stochastic model, the overall statistics of the overdeeping features would be reasonably well represented, but their geographical locations would not.

With regard to confluences, and following Goff and Nordfjord (2004), where single channel elevation surfaces overlap we accept the maximum depth. This introduces a hierarchical element to surface prediction, whereby deeper channels are favoured

over shallower ones. However, as the approach is based solely on topography (not rock type or age as such information are rarely available), this introduces a limitation that cannot easily be resolved in light of such sparse observations. We suggest that, in the absence of data, use of the deepest value is preferable over shallower values, due to the overall systematic overes­timation of bed-elevation (i.e. underestimation of depth) by Bed2013 (see Sect 4.1). The presence of overdeepenings within glacial environments is well established (c.f. Cook and Swift, 2012), their distribution having been observed from bed DEMs

for beneath contemporary ice sheets including Greenland (Patton et al., 2016). However, there remains limited quantitative



data on their morphology with which to understand the processes of their development (Patton et al., 2015) and the specific relationship between fjord network structure and the locations of overdeepenings and the sills between them. Should additional information become available, such an approach to establish their location could be implemented by introducing rules - for example, *an overdeepening of a given step lowering occurs where two fjords of a given width and known depth confluence.*

Another approach would be to develop a set of rules which incorporate a fjord hierarchy akin to stream order and their associated Strahler numbers (Strahler, 1957).

The majority of end users of a new Greenland bed DEM including improved bathymetry are expected to be within the ice sheet and polar ocean modelling communities. With this in mind, the approach presented here has been tailored to best suit

the purpose of end products that have fjord bathymetry constrained by the synthetic algorithm. Since the algorithm performs better closer to the glacier termini, as opposed to the fjord mouth, users of DEM products based upon this algorithm would be encouraged to focus on processes from the glacier-to-fjord direction (e.g. calving) as opposed to processes focused from the fjord-to-glacier direction (e.g. ocean forcing as in Murray et al., 2010). The impact of high and low frequency stochastic perturbations for topographic datasets for ice sheet modelling is well documented, with models being more sensitive to spatially

broad low frequency noise as opposed to higher frequency noise of the same magnitude (Sun et al., 2014). To predict the precise geographical location of sills and overdeepenings with the limited information known for many fjords is a near impossible task. However, as described in the previous paragraph, the statistical features of these features could be represented by a stochastic model. To the best of our knowledge, our study is the first to consider the statistical properties of fjord bathymetry. This is a significant development as constraining models of high frequency are important where bathymetric surfaces are used to mimic

calving (e.g. Lee et al., 2015) or spinning-up ice sheet models over larger regions (e.g. Bindschadler et al., 2013). The exponent for the high frequency tail of the fjord bathymetry power spectrum, 1.74, is consistent with other exponent values found for seafloor topography (Bell, 1975; Goff and Jordan, 1988) and serves as a preliminary guide for future stochastic models. The transition wavelength ($\sim$ 15 km) for the fjord power spectra is shorter than for abyssal hill features in the mid ocean, where the wavelength value is $\sim$ 40 km (Bell, 1975).

## 6 Summary

Until now, bed-bathymetry DEMs for coastal regions of Greenland have been limited by sparse observations and simplistic interpolation methods being applied within fjord regions. The presented algorithm for synthetic fjord bathymetry provides a new approach to generate bathymetric geometry along fjords. The method takes advantage of observational data where available and

assumes that fjords maintain a parabolic cross-sectional profile, thus capturing a leading-order geometric constraint from the ice flow geomorphological processes largely responsible for fjord formation. The validity of the algorithm was tested through comparison with new observational bathymetry data for a fjord system in North West Greenland, and better overall agreement with the data was observed than with Bed2013. Additionally, we performed an initial assessment for how a stochastic model

of fjord bathymetry could be parameterised, and thus how high frequency perturbations to the flat synthetic geometry could be modelled. The physical validity of the algorithm is limited at multiple channel confluences, as the hierarchy of processes responsible for the landscape features is not explicitly incorporated in the algorithm.

Until more observational data are available, this algorithm provides a suitable estimate for simulating previously unmapped fjord geometry. The presented method will be used to assist in the mapping of fjords within the next Greenland bed DEM data product and has potential application for Antarctica. Using the results of the stochastic model analysis, multiple Greenland bed DEM realisations will be produced, offering the opportunity for the running of ensemble ice sheet model simulations. The release of this new dataset is proposed for 2017.

**Data Availability**

All data used for the preparation of this manuscript are openly available. The GIMP land classification mask is available and fully documented in Howat et al. (2014). Bed2013 is available and fully documented in Bamber et al. (2013). The IBCAO (v3) DEM is available and fully documented in Jakobsson et al. (2012). The OBS1516 dataset was compiled by I. Fenty and was constructed from (1) the OMG and (2) the Ummannaq and Viagat fjord system bathymetric datasets which are documented and available from OMG Mission (2016) and Rignot et al. (2016) respectively.

**Author contribution**

CNW, TMJ, JAD, MJS and JLB were involved in the development of the overall methodological framework and interpreted the results. SLC and CNW developed the code to map fjord centrelines. CNW, CDC, DAS and AS were involved in discussions with regard to the introduction of fjord shape and overdeepenings. TMJ implemented the processing of the spectral analysis of the fjord profiles. IF provided the OBS1516 dataset. CNW wrote the manuscript with comments and contributions from all
other authors.

**Conflict of interest**

JLB is a member of the editorial board of the journal. All other authors declare that they have no conflict of interest.

*Acknowledgements.* This study was supported by UK NERC grant NE/M000869/1.



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





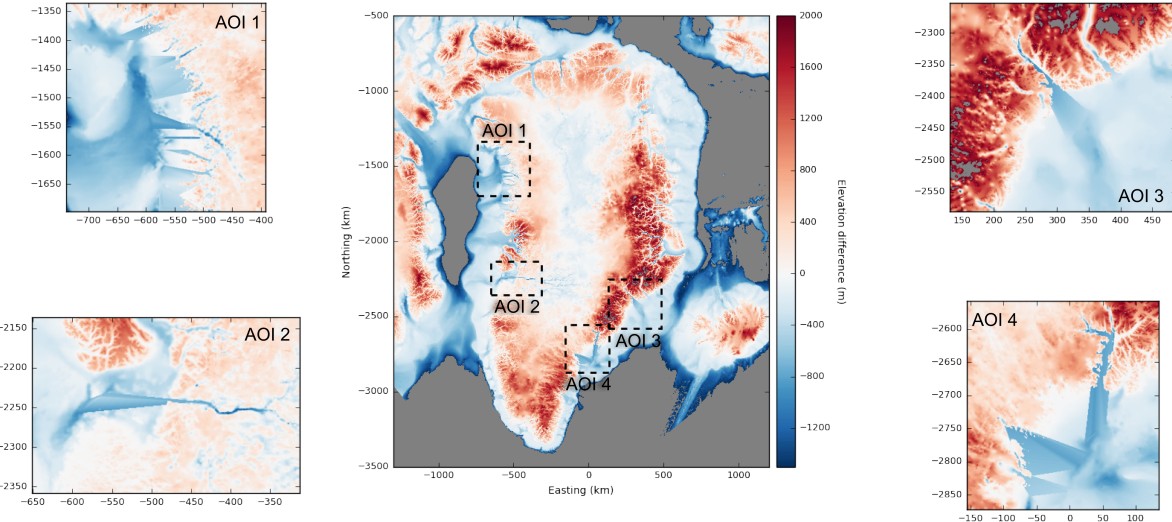

**Figure 1.** Examples of non-physical bathymetry around the coast of Greenland following Bamber et al. (2013), using only observations included within the IBCAO v3 (Jakobsson et al., 2012) DEM. Within the fjord mouths, discontinuities in the direction of ice flow were removed, resulting in discontinuities at the lateral boundaries.

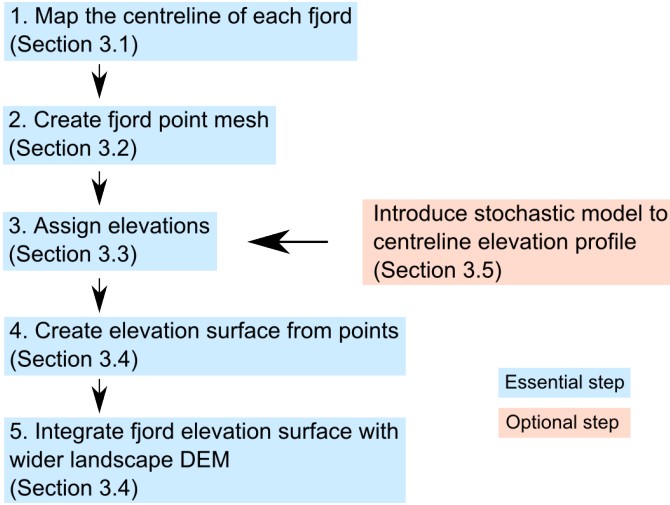

**Figure 2.** Algorithm flow diagram.

Vieli, A. and Nick, F.: Understanding and Modelling Rapid Dynamic Changes of Tidewater Outlet Glaciers: Issues and Implications, Surveys in Geophysics, 32, 437–458, doi:10.1007/s10712-011-9132-4, 2011.

Wheeler, D.: Using parabolas to describe the cross-sections of glaciated valleys, Earth Surface Processes and Landforms, 9, 391–394, doi:10.1002/esp.3290090412, 1984.



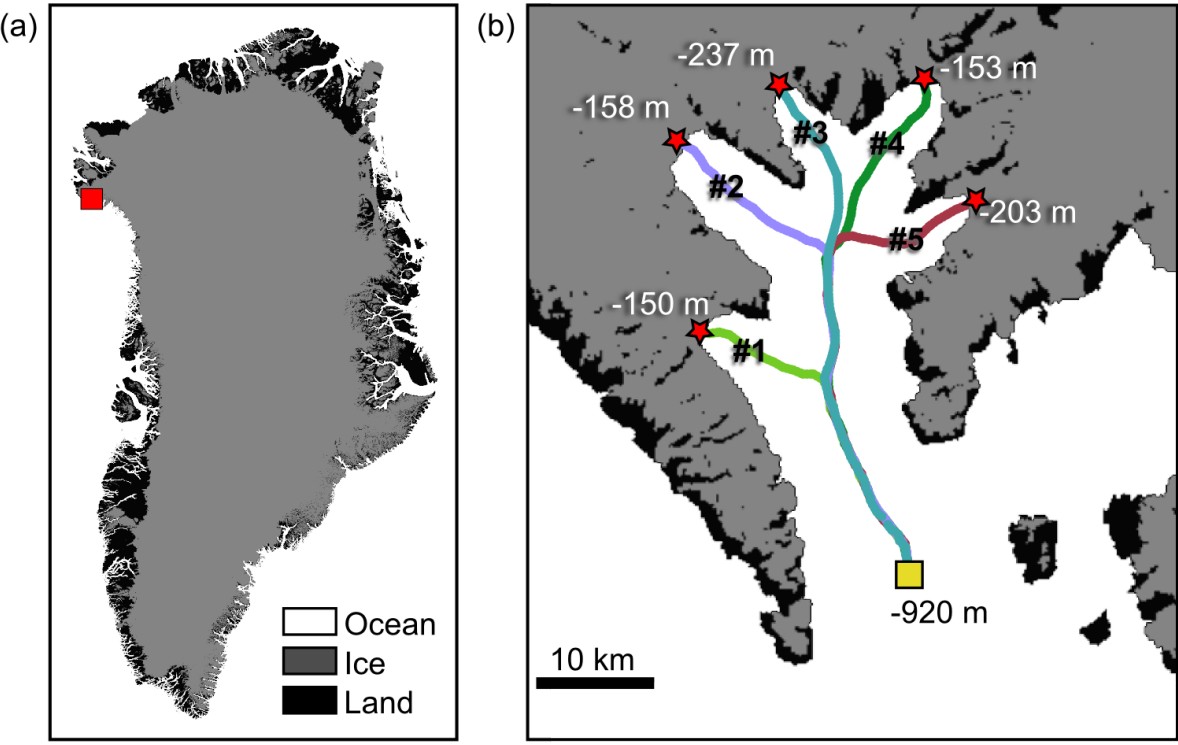

**Figure 3.** (a) The Cape York area of interest in North West Greenland (red box - zoom displayed in (b)) relative to the Greenland land classification mask (Morlighem et al., 2014) (b) The fjord system area of interest with mapped channel centrelines and seed (red star) and mouth (yellow square) elevations as identified from observations (Rignot et al. 2016, OMG Mission 2016).





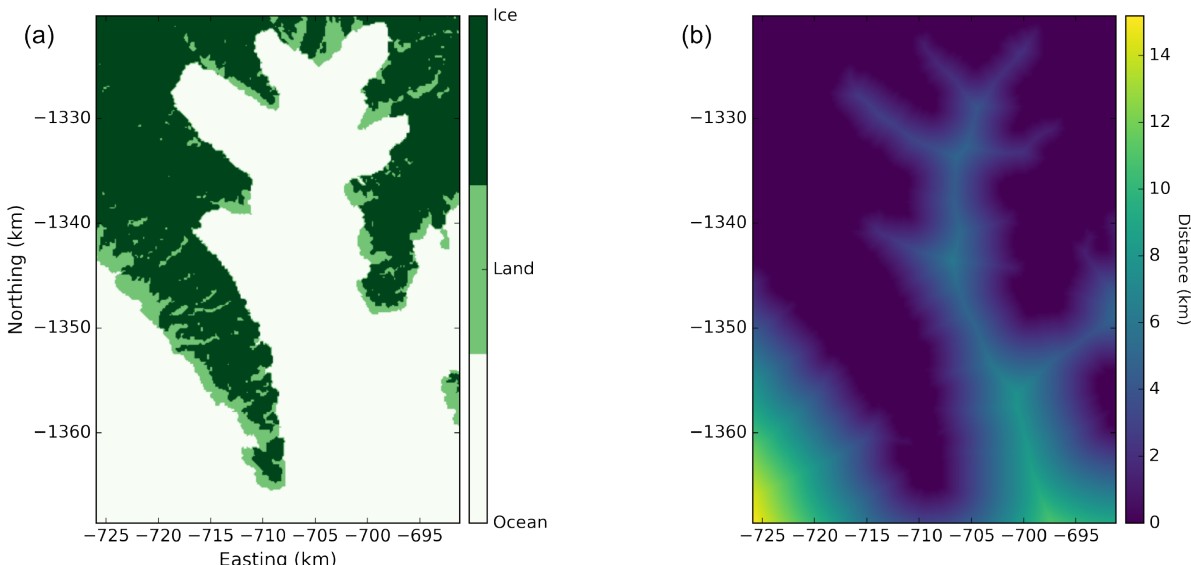

**Figure 4.** The area of interest close to Cape York illustrating (a) land classification taken from the GIMP land mask (Howat et al., 2014; Morlighem et al., 2014) and (b) the distance of ocean regions relative to land/ice regions.



**Figure 5.** Fjord centreline approach where (a) fjords are identified from the GIMP land classification mask (Morlighem et al., 2014) as being sections of ocean between land/ice regions (b) An example of the centreline pathfinder algorithm as applied to a single fjord. From a seed point, children are spawned, each time resulting in the creation of new branches. Where children intersect the land/ice boundary, branches are culled. The culling of branches 2.1.1 and 2.1.2 within the pink box, as well as the selection of the shortest path is discussed in the text. Please refer to the online version of this article to make use of references to colour.





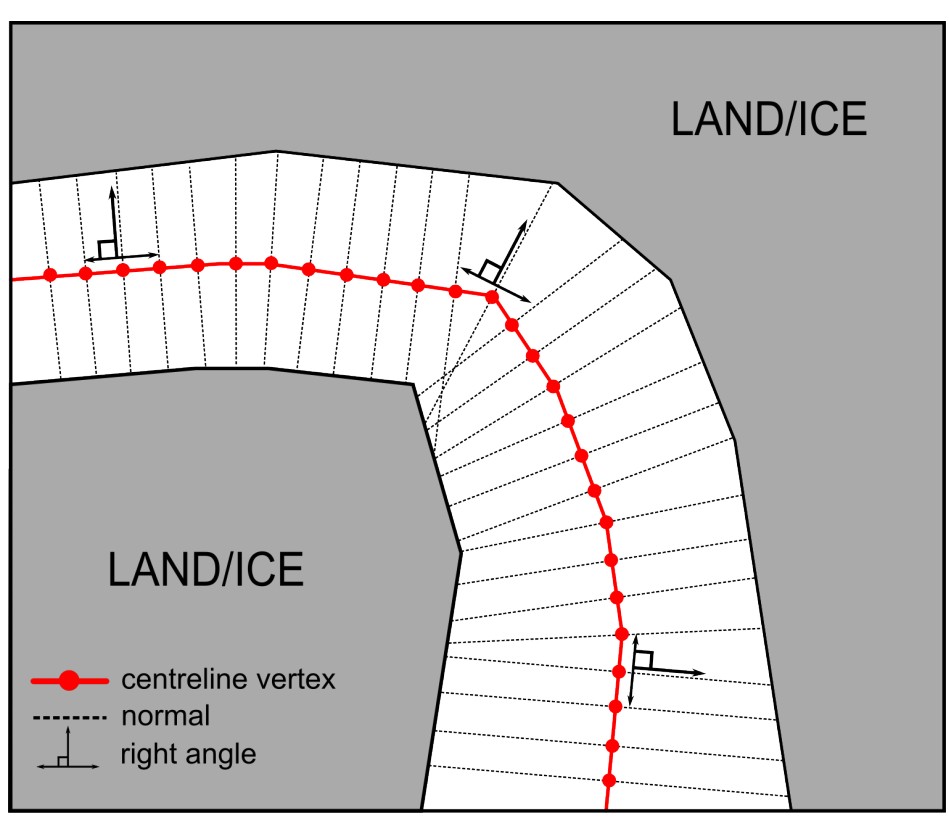

**Figure 6.** For a given centreline vertex, new points (black dashed lines) are created normal to the centreline trajectory (solid red line), up to the sides of the channel as defined by the land/ocean mask (Howat et al., 2014; Morlighem et al., 2014). Normal angles are calculated relative to the vector between the neighbouring points of a given vertex.



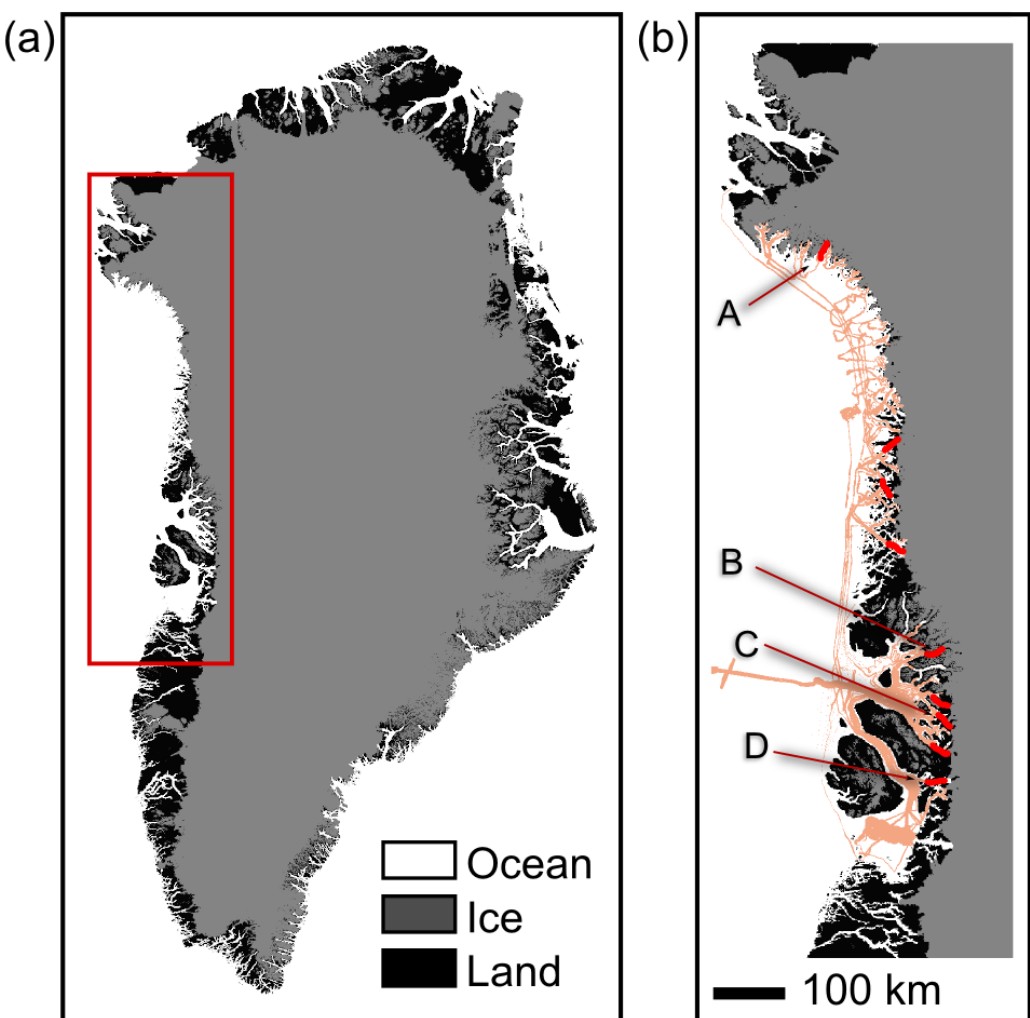

**Figure 7.** (a) Area of interest considered for selection of fjord centrelines for spectral analysis relative to the GIMP land classification mask (Morlighem et al., 2014) (b) Fjord centrelines (red) selected along the west coast of Greenland, with the bathymetric observational DEM also displayed (in orange) (Rignot et al. 2016, OMG Mission 2016) — centrelines were only selected where data were available with gaps affecting < 20 % of the overall profile length. Labels on (b) relate to the profiles illustrated in Fig. 8.



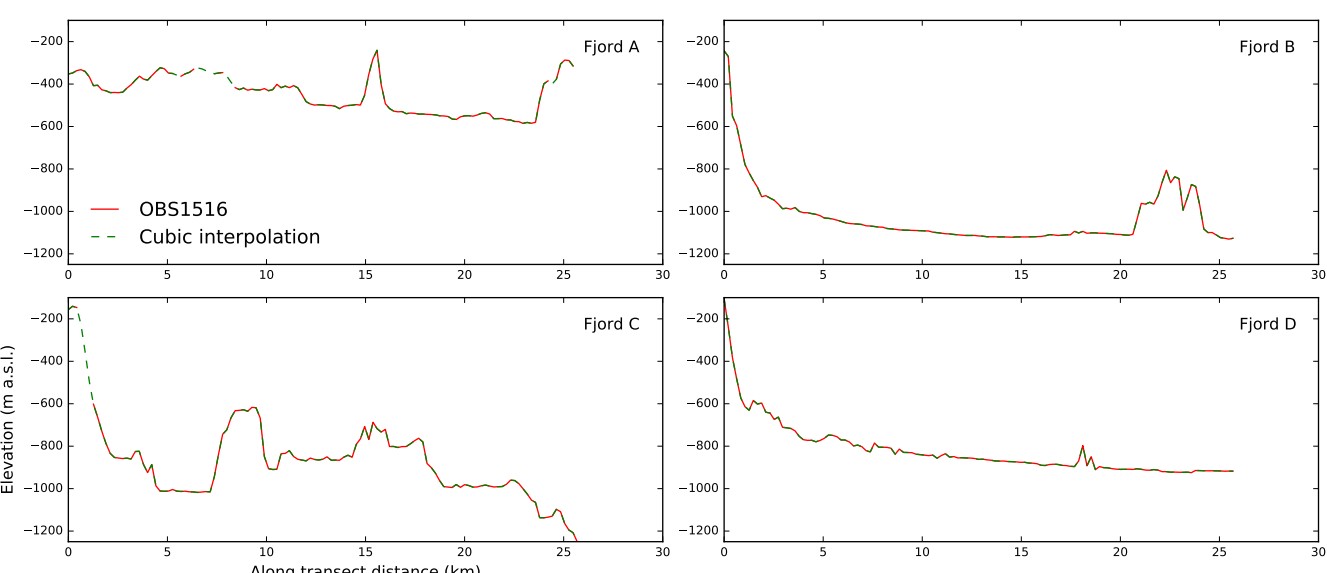

**Figure 8.** Examples of the along transect profile of four fjords from the area of interest depicted in Fig. 7. Along transect distance starts at the head of each fjord and extends to the fjord mouth.





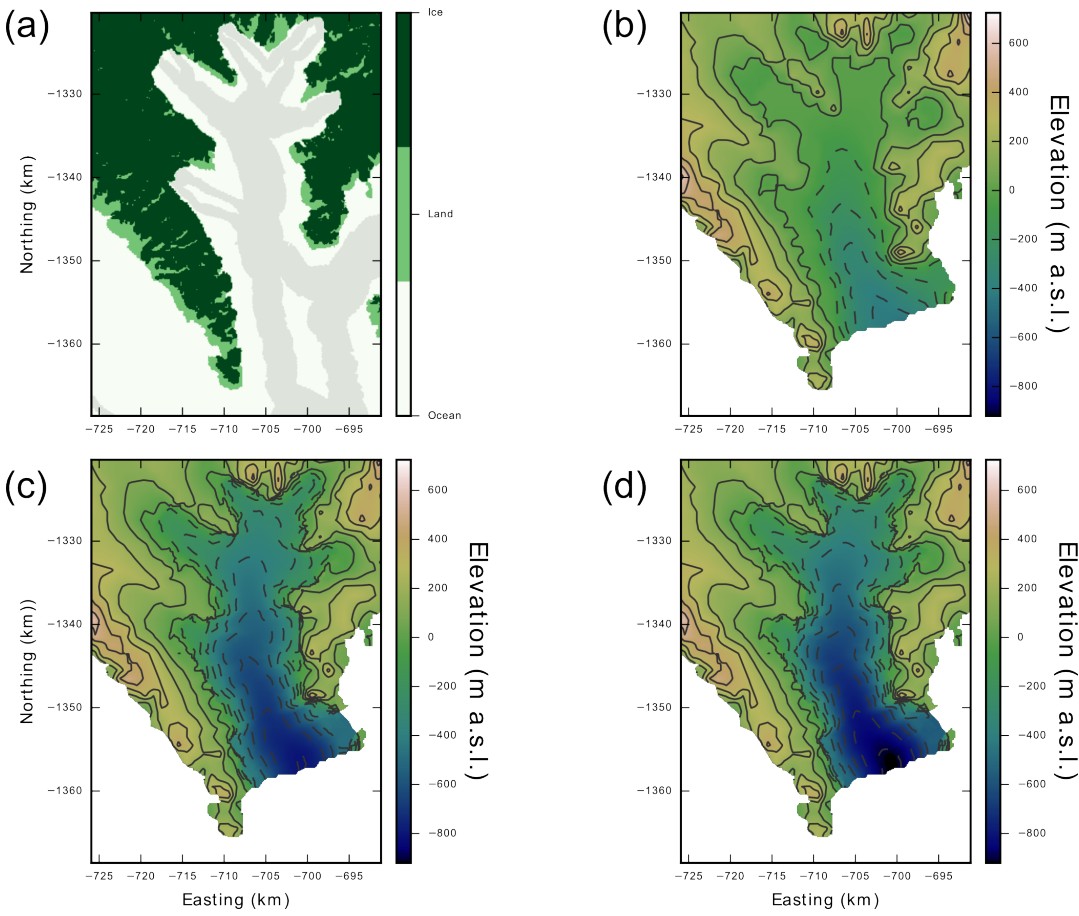

**Figure 9.** The area of interst close to Cape York displaying (a) the land mask of the area of interest (Morlighem et al., 2014) with the extent of OBS1516 in grey, (b) Bed2013 elevation, (c) Bed2013 combined with the SynthIBCAO synthetic geometry and (d) Bed2013 with the inclusion of the SynthOBS synthetic geometry. SynthIBCAO and SynthOBS are only used within the ocean regions of the land mask.





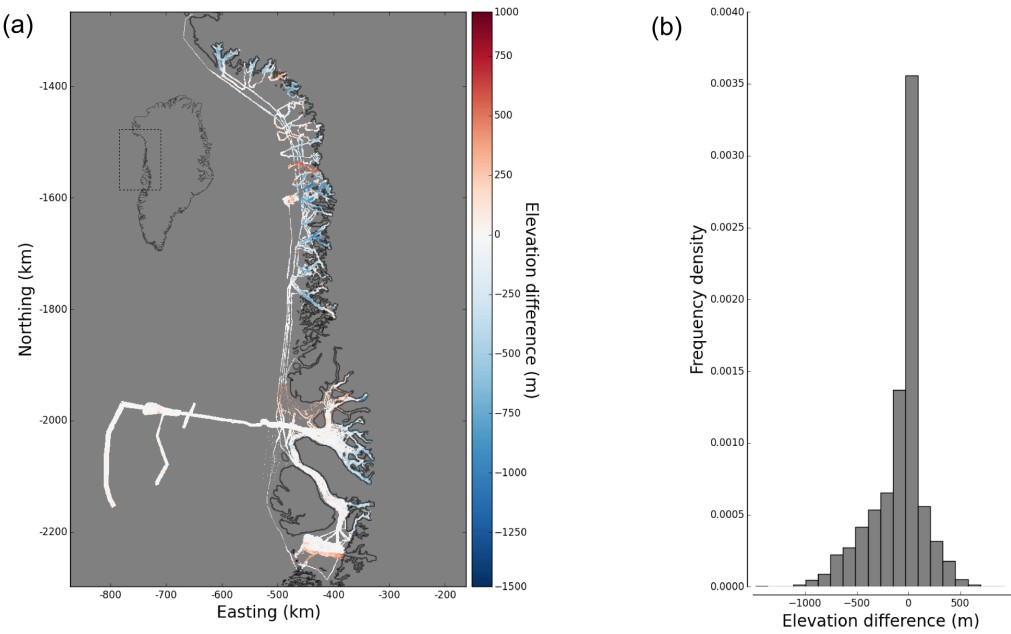

**Figure 10.** Bed elevation differences between Bed2013 and OBS1516 at all surveyed locations along the west Greenland coast in (a) plan view and (b) as a histogram. Red regions in (a) indicate bathymetry elevation underestimation by Bed2013 (not deep enough), with blue regions illustrating overestimation (too deep).





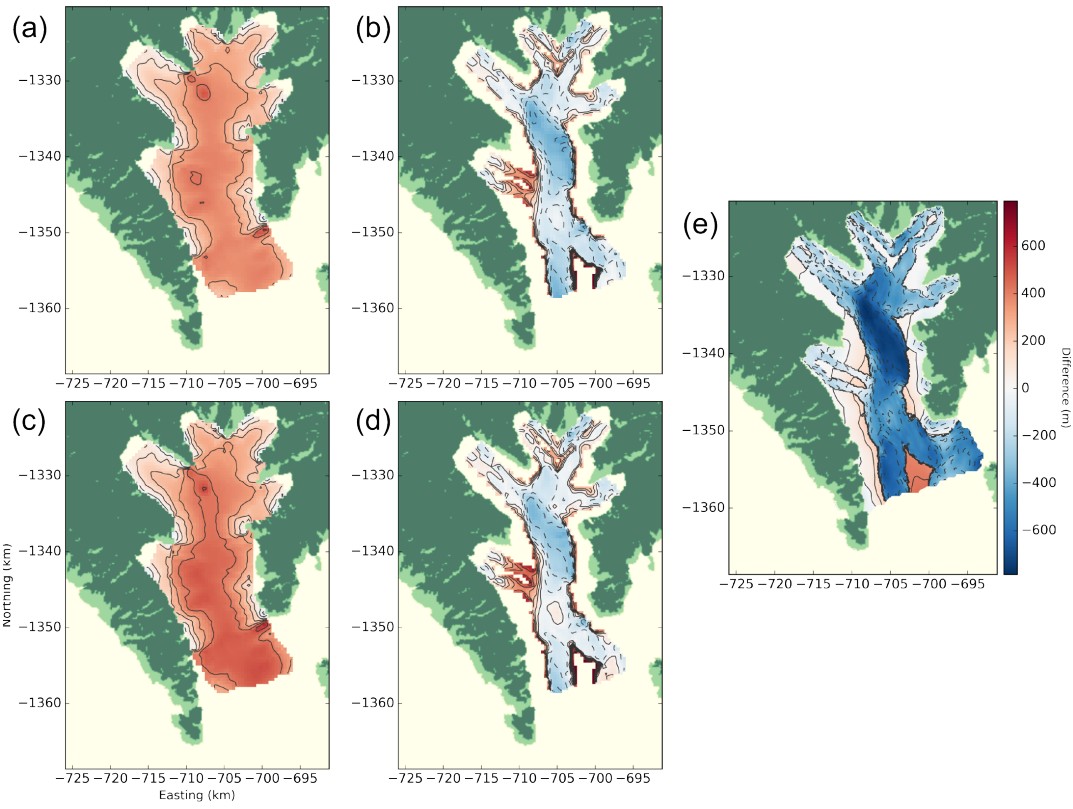

**Figure 11.** Bed elevation differences displaying (a) Bed2013 minus SynthIBCAO, (b) OBS1516 minus SynthIBCAO, (c) Bed2013 minus SynthOBS, (d) OBS1516 minus SynthOBS and (e) OBS1516 minus Bed2013 within the Cape York area of interest. Red illustrates where the subtrahend is deeper than the minuend - in the case of OBS156 minus a surface, this indicates an elevation overestimation (too deep).





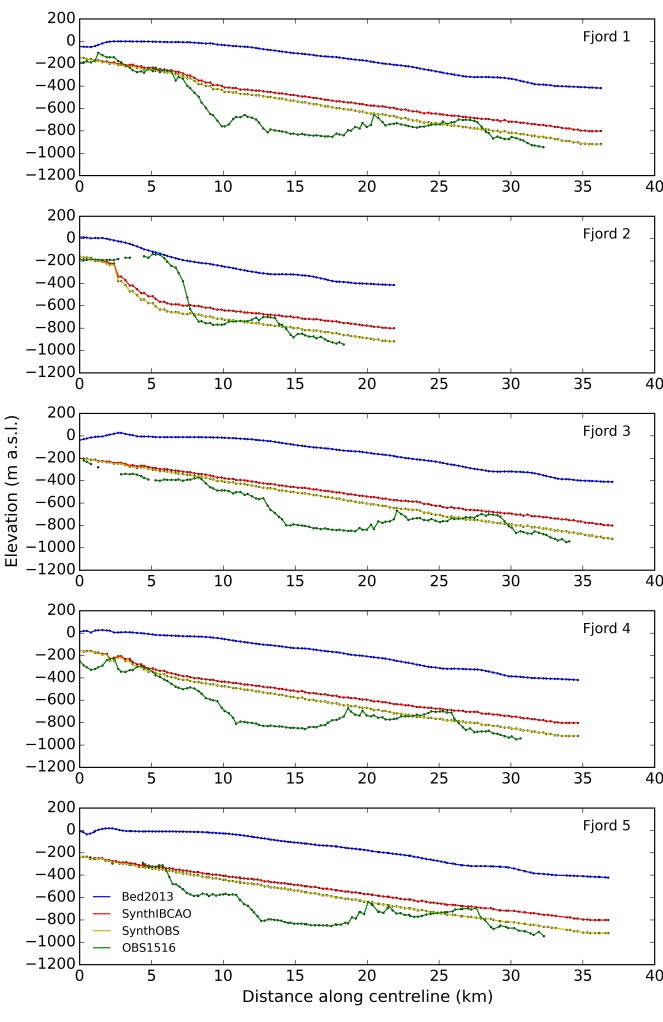

**Figure 12.** Centreline elevation profiles from Bed2013, OBS1516 and the SynthIBCAO and SynthOBS synthetic algorithm approaches. All profiles extend from the head of each fjord to the mouth as depicted in Fig. 3.





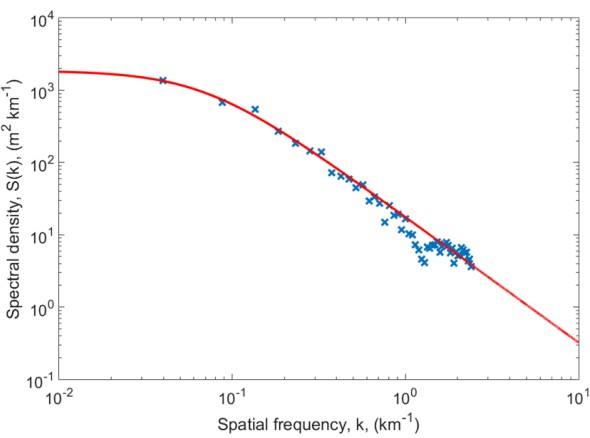

**Figure 13.** Composite power spectral density from fjord bed elevation profiles. Blue crosses indicate spectral data that has been averaged over nine fjord profiles, and the solid red curve is the best fit to the parametric model, equation 2.





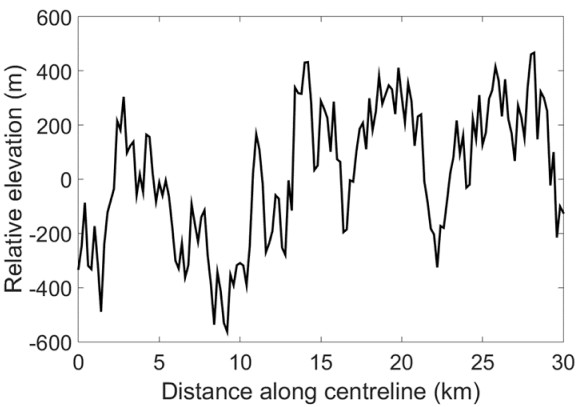

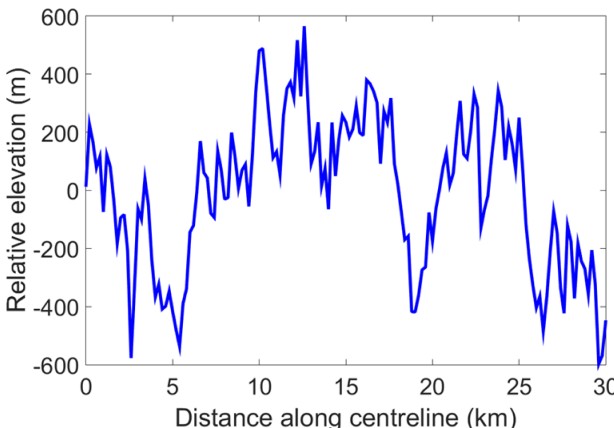

**Figure 14.** Two different realisations of the stochastic model for high frequency perturbations to the synthetic fjord elevation profiles. The model uses the parametric fit in Fig. 13 to generate the profiles.