# Peer review of "Generating synthetic fjord bathymetry for coastal Greenland"

_The Cryosphere, 2016_

## Referee Comment (RC1) · J. Goff (Referee) · 11 Oct 2016

General Comments: Overall this is a good paper worthy of publication with just minor revisions. The methodology described here should find some important uses in glacier modeling studies by greatly improving the predicted local bathymetry within fjords compared to the regional interpolations. The text is clear, well written, well organized, and the figures are very good. I do have a few suggestions for improvements.

Specific Comments: 1) Page 2, line 16: the term "physically based" is too vague.

2) Page 3, Line 19: Goff et al. (2014) should be referenced with regard to statistical modeling and interpolation.

3) Page 5, line 30: It is stated that "to the best of our knowledge, there are no studies

that have considered spectral analysis of fjord bathymetry." That is technically true, but Goff et al. (2014) certainly investigated the statistical characteristics of subglacial channel geometry in great detail, and that has to be considered highly relevant to this study.

4) Page 7: This description of the methodology for centerline picking includes a number of seemingly arbitrary values: an unknown "predefined distance interval", parent edge < pi/6, angle between any pair of new edges no less than pi/24, |x-xi|, |y-yi| < 16 km, |a-ak| < pi/8. The authors should endeavor to provide rationales for these values.

5) Pages 7,8: I think that the demonstration of centerline picking would greatly benefit from using a path that was not a perfect straight line. This part is a bit hard to follow, and I think a curved path would give the reader a more intuitive sense about how it really works. I would suggest in particular that the geometry used for Figure 6 be the same as used for Figure 5. I also wonder: in looking at Figure 4b, wouldn't it be a lot simpler to just follow the path of maximum distance?

6) Page 14, line 26: It is inaccurate to say that this is the first time such a methodology has been applied to fjords since subglacial channels are geometrically identical to fjords.

---

## Referee Comment (RC2) · Anonymous Referee #2 · 4 Nov 2016

This paper presents a novel method for generating synthetic bathymetry for fjords in Greenland - areas where previous datasets produce somewhat spurious features due to a lack of data to constrain them and an unsuitable interpolation method. 'Sensible' bathymetry is of great value to glaciological and oceanographic studies in these regions, even in the absence of smaller scale variations. The paper also provides a valuable starting point for imposing stochastic variations to represent these smaller scale variations. The method is well thought through, and the paper is well written and structured, and clear to follow. As such I recommend the paper for publication with a few minor changes.

I agree with John Goff that the description of the centreline method could be made clearer if Figures 5 & 6 were comparable, and if Fig.5 included a bend. Could you have Fig. 5a as a straight fjord to clearly demonstrate the algorithm, and then a 5b with a

simple bend? Then you could probably leave Fig. 6 as it is.

I found it hard to compare Figs. 12 & 14 because of the different sizes of the plots and axis scales. I know they aren't displaying exactly the same thing, but it would help the visual analysis if the x and y axes were equivalent in scale.

Two small typos:

p7, line 24: 'manner' not 'manor'

p10, line 7: should be 'analysis' rather than 'analyses'?

---

## Referee Comment (RC3) · R. Timmermann (Referee) · 15 Nov 2016

Reviewer's comment on

"Generating synthetic fjord bathymetry for coastal Greenland"

by Christopher N. Williams and coworkers

submitted to The Cryosphere Discussions

**General comments**

In this paper, the authors present a method for generating synthetic, but realistic fjord bathymetries in a scarce-data environment, where "realistic" means that topography represents many important aspects of reality even if it cannot reflect the true situation exactly. Improvements compared to previous methods are shown and demonstrate the ability of the method. A pan-Greenland application is envisaged.

The paper makes a reviewer's life very easy. The topic is interesting, the method obviously very useful, results are meaningful. Structure of the manuscript is clear, figures are useful, presentation in general is very pleasant. I particularly liked the review of past approaches to synthetic channel geometries in (not only) Greenland DEMs. I have a couple of specific comments that may help to improve the manuscript even further. I recommend to accept the paper for publication, encouraging minor revisions.

**Specific Comments**

1. page 2, l. 5-6: The "physically unrealistic morphologies" in the Bamber et al. (2013) dataset have also been recognized by Schaffer et al. (2016) when putting together RTopo-2 – although the approach to overcome the issues there was much more heuristic than what is presented in this paper. Still, you may want to add the Schaffer et al. (2016) work to the list of studies aiming for an improved representation of Greenland fjord topography in a bed-to-bathymetry DEM.

2. page 3, l. 9: I couldn't do much with the word "acknowledged" here.

3. page 3, l. 25: I suspect that the "and," at the end of the line was not intended to be there. I think this can nicely be two separate sentences.

4. page 4, l. 18: I suggest to make it "low-resolution" (because it makes it easier to see that the "large" belongs to "datasets", not to "resolution")

5. page 5, l. l 27: "a flat spectra" sounds wrong to me

6. page 7, l. 5: It's actually a path integral of a constant number (most likely One), isn't it? "Along-track distance"or just "length" would be simple words for it.

7. page 7, l. 16: I'll have to admit that I could not do much with the word "transform" here.

8.  page 7, l. 24: manor -> manner

9.  page 7, l. 30-34: It is clear that some of the potential paths have to be removed along the track, but I'll have to admit I fail to understand the explanation why step 4 does this job the way it is supposed to be. Any chance to put this into some simple descriptive words illustrating the idea and the reasoning?

10. page 8, l. 7: I think the explanation for step 7 is easier to understand if the words "When considering the length of all complete paths," are removed and the sentence simply starts with "Where …".

11. page 9, l. 17: I think there should be no comma after "parameters".

12. page 13, l. 22: I suggest to remove the paragraph break here. Maybe the sentence in l. 20-22 can be rewritten to clarify that the improvement is an improvement compared to Bed2013, with OBS1516 as a reference what the targeted  truth is.

13. page 14., l.1: I think "provided" or "presented" is better here than "illustrated"

14. page 15, l. 10: "_at_ a significant distance" ?

15. page 16, l. 21: "high-frequency" ?

16. page 16, l. 34: should it be "_for_ Bed2013" ?

17. caption to Figure 5: Even with the electronic version in front of me, the "Please refer to the online version of this article to make use of references to colour." bit fails to make sense to me.

18. caption to Figure 8: I suggest to make it "along-transect" at both locations

19. Figure 9: The figure shows that SynthIBCAO and SynthOBS give very similar results, which is good and demonstrates the ability of the method. However, I miss a possibility to directly compare to the OBS1516 data. Would it be possible to replace the grey area in panel a, which just indicates data coverage, by the OBS1516 elevation data? Keeping the extent / data coverage information of course (white areas remain white).

20. caption to Figure 11: I am sure the last sentence can be formulated in an easier way (still making the point to be made)

21. In general, I miss a statement (e.g. in the Summary) on how many manual steps are required during the procedure. Manual steps are mentioned at several locations in the manuscript, but I wonder how much work it would be to do this for, let's say, entire Greenland.

14.11.2016, Ralph Timmermann

---

## Author Comment (AC1) · 21 Dec 2016

**Author comments: Generating synthetic fjord bathymetry for coastal Greenland**

**C. N. Williams et al., The Cryosphere**

**Review by: J. Goff**

We thank the reviewer for their supportive comments and have made corrections in line with the provided suggestions, much improving the original manuscript.

*1) Page 2, line 16: the term "physically based" is too vague*

> 'Physically based' had been used to differentiate between purely statistically calculated geometry as opposed to a geometry specifically implemented to be representative of "real" geomorphology or bathymetry. We agree that the term used is non-specific and therefore to reduce vagueness, we have replaced it with "geomorphologically realistic" at lines 16 (pg 2), 4 (pg 6), 30 (pg 14). The reference at line 31 (pg 4) has been removed.

*2) Page 3, Line 19: Goff et al. (2014) should be referenced with regard to statistical modeling and interpolation.*

> The reference has now been added.

*3) Page 5, line 30: It is stated that "to the best of our knowledge, there are no studies that have considered spectral analysis of fjord bathymetry." That is technically true, but Goff et al. (2014) certainly investigated the statistical characteristics of subglacial channel geometry in great detail, and that has to be considered highly relevant to this study.*

> We have now deleted line 30 and replaced it with:
>
> "Whilst the spectral properties of mid-ocean bathymetry (Bell 1975, Goff and Jordan 1988) and subglacial channels (Goff et al. 2014) have been assessed, to the best of our knowledge this has not been done for fjord bathymetry."

*4) Page 7: This description of the methodology for centerline picking includes a number of seemingly arbitrary values: an unknown "predefined distance interval", parent edge < pi/6, angle between any pair of new edges no less than pi/24, |x-xi|, |y-yi| < 16 km, |a-ak| < pi/8. The authors should endeavour to provide rationales for these values.*

> The referee is correct to note that these parameters are somewhat arbitrary. Ultimately, they are chosen so that at least one path is found for each fjord, but not many more. We plan to formalise this process for the final DEM product
>
> Essentially, what we are doing here is solving a optimization problem many times to find the optimal paths between a large set of start- and end-points, then choosing a subset of the them such that there is only one path between each pair of close start and end points. We have complicated the process somewhat by attempting to carry out the path generation and selection at the same time, in order to avoid the large number of paths that we would otherwise need to consider.
>
> The predefined distance interval is a simple finite difference parameter, and is picked so that there are enough nodes on a path to resolve it – this would be the same however we implemented the above. The ideal (but unattainable) value would be zero.

The relationship between child and parent edges is chosen to be small enough allow the path to turn quickly enough to follow the channel, and large enough so that the minimum radius of curvature is of the order of one channel width. The ideal value would be 2*pi, and restricting it simply anticipates the expectation that paths containing many loops will ultimately be rejected. In other words, this keeps paths close to the locally optimal direction, but allows some latitude so that branches can form.

The angle between new paths (pi/24) is another finite difference parameter – we cannot consider the continuum of angles spanning (-pi/6,pi/6). Again the ideal value would be zero and we reduce it progressively until the same (or similar enough) set of paths are generated.

The values |x-xi|, |y-yi| < 16 km are chosen to identify similar paths. The first two are chosen such that one of a set of seeds is identified within a small number of generations, if the paths that start from from them appear to be converging.

The condition |a-ak| < pi/8 is also chosen to identify similar paths. The angle must be greater than zero to allow branches to form from paths which are identical up to the point of the branch, and persist provided they arrive at distinct end point. A smaller angle reduces the number of branches in play at any one generation.

We have added additional text to section 3.1 to add further clarity.

*5) Pages 7,8: I think that the demonstration of centerline picking would greatly benefit from using a path that was not a perfect straight line. This part is a bit hard to follow, and I think a curved path would give the reader a more intuitive sense about how it really works. I would suggest in particular that the geometry used for Figure 6 be the same as used for Figure 5. I also wonder: in looking at Figure 4b, wouldn't it be a lot simpler to just follow the path of maximum distance?*

In line with the suggestions of both reviewers 1 and 2, to make the methodology as clear as possible, we have amended figure 5 so that it uses the same geometry as presented in figure 6. The algorithm is based on following the path of maximum distance, and all the additional complexity comes about because there are many possible start and end points. A simple approach might be to compute every optimal path between every possible start and end, and then choose among them somehow. For example, imagine two possible start points (seeds) and one possible end point. There are two optimal paths, and we need some way to decide whether to keep both, or just one (e.g if they are at the start of the same fjord). Our method is really just attempting to do this choosing 'on the fly', by discarding paths that start at nearby points to a 'better' path if they appear similar otherwise (have the same centroid and direction). By 'better' we mean that a path is closer to the centre of the channel (as measured by the maximum distance) on average along its length.

We have added additional text to section 3.1 to add further clarity.

*6) Page 14, line 26: It is inaccurate to say that this is the first time such a methodology has been applied to fjords since subglacial channels are geometrically identical to fjords.*

We agree with the reviewer and have amended the top of the discussion. The discussion now begins **"Channel elevation point meshes have been implemented in different research fields including hydrology (Merwade et al., 2005, 2008), and glaciology (Goff et al., 2014). This study provides a key addition, which addresses sparse data availability with…"**

---

## Author Comment (AC2) · 21 Dec 2016

**Author comments: Generating synthetic fjord bathymetry for coastal Greenland**

**C. N. Williams et al., The Cryosphere**

**Review by: Anonymous**

We thank the reviewer for their supportive comments and have made corrections in line with the suggested improvements to the manuscript.

1. I agree with John Goff that the description of the centreline method could be made clearer if Figures 5 & 6 were comparable, and if Fig.5 included a bend. Could you have Fig. 5a as a straight fjord to clearly demonstrate the algorithm, and then a 5b with a simple bend? Then you could probably leave Fig. 6 as it is.

   We agree with both reviewers 1 and 2 on this point and have amended Fig. 5 so that (5a) now represents that which was Fig. 5(b) in the original manuscript. The new 5(b) now uses the same geometry as presented in Fig. 6.

2. I found it hard to compare Figs. 12 & 14 because of the different sizes of the plots and axis scales. I know they aren't displaying exactly the same thing, but it would help the visual analysis if the x and y axes were equivalent in scale.

   We agree with the reviewer. This is a good point and we have now changed the width of Figure 14 (the high frequency stochastic model) to better match Fig. 12 (centreline elevation profiles). When doing this we realised that our stochastic model amplitudes were not correctly normalised, so we have re-done the plot 14 with correctly normalised amplitudes. The new, correctly normalised amplitudes appear significantly smaller than the real fjord elevation range (since they just deal with the higher frequency perturbation). In order to make this point clear, and to better relate Figs 12 and 14 we have modified caption 14 as follows:

   Figure 14: Two different realisations for the stochastic model for high frequency perturbations to the synthetic fjord elevation profiles. The model uses the parametric fit in Fig. 13 to generate the profiles, and is statistically consistent with the OBS1516 bathymetric profiles (green lines in Fig. 12). The overall trend of the fjord bathymetry and lower frequency oscillations (corresponding to wavelengths ~14 km or greater) are not synthetically generated and explains why the amplitude of the modelled elevation) is significantly less than the bathymetric observations in Fig 12.

3. p7, line 24: 'manner' not 'manor'

   Done.

4. p10, line 7: should be 'analysis' rather than 'analyses'?

   Done.

---

## Author Comment (AC3) · 21 Dec 2016

**Author comments: Generating synthetic fjord bathymetry for coastal Greenland**

**C. N. Williams et al., The Cryosphere**

**Review by: R. Timmermann**

We thank the reviewer for their supportive comments and have made corrections in line with the provided suggestions, much improving the original manuscript.

1. *page 2, l. 5-6: The "physically unrealistic morphologies" in the Bamber et al. (2013) dataset have also been recognized by Schaffer et al. (2016) when putting together RTopo-2 – although the approach to overcome the issues there was much more heuristic than what is presented in this paper. Still, you may want to add the Schaffer et al. (2016) work to the list of studies aiming for an improved representation of Greenland fjord topography in a bed-to-bathymetry DEM.*

   This new compilation is now acknowledged in the new manuscript. We now include:

   "…towards the ice sheet margins. The development of the RTopo-2 provides another response to the limitations of Bed2013 within fjord regions, with improvements being made by including new observational data (Schaffer et al., 2016)"

2. *page 3, l. 9: I couldn't do much with the word "acknowledged" here.*

   We have removed "acknowledged" from the sentence.

3. *page 3, l. 25: I suspect that the "and," at the end of the line was not intended to be there. I think this can nicely be two separate sentences.*

   This now reads: "…variogram function (Deutsch and Journel, 1998). Using this it is possible…"

4. *page 4, l. 18: I suggest to make it "low-resolution" (because it makes it easier to see that the "large" belongs to "datasets", not to "resolution")*

   We agree. This has now been altered to read: "…are apparent in low resolution datasets particularly…"

5. *page 5, l. l 27: "a flat spectra" sounds wrong to me*

   By 'flat spectra' we simply mean that there is a regime where that the power spectral density is (near) independent of frequency. These term is in common usage in signal processing, and we see no problem with the usage here. However, for clarity we now say: "flat region of the power spectra"

6. *page 7, l. 5: It's actually a path integral of a constant number (most likely One), isn't it? "Along-track distance"or just "length" would be simple words for it.*

   We now use the term "path length" instead of "path integral".

7. *page 7, l. 16: I'll have to admit that I could not do much with the word "transform" here.*

   This has now been rewritten to read: "…we calculate the distance of all locations between land/ice and ocean within the channel (d), from which the…"

8. *page 7, l. 24: manor -> manner*

   Done.

9. *page 7, l. 30-34: It is clear that some of the potential paths have to be removed along the track, but I'll have to admit I fail to understand the explanation why step 4 does this job the way it is supposed to be. Any chance to put this into some simple descriptive words illustrating the idea and the reasoning?*

   The lack of transparency of this step was also highlighted by reviewer 1, to whom we responded with the following:

   The referee is correct to note that these parameters are somewhat arbitrary. Ultimately, they are chosen so that at least one path is found for each fjord, but not many more. We plan to formalise this process for the final DEM product

   Essentially, what we are doing here is solving a optimization problem many times to find the optimal paths between a large set of start- and end-points, then choosing a subset of the them such that there is only one path between each pair of close start and end points. We have complicated the process somewhat by attempting to carry out the path generation and selection at the same time, in order to avoid the large number of paths that we would otherwise need to consider.

   The predefined distance interval is a simple finite difference parameter, and is picked so that there are enough nodes on a path to resolve it – this would be the same however we implemented the above. The ideal (but unattainable) value would be zero.

   The relationship between child and parent edges is chosen to be small enough allow the path to turn quickly enough to follow the channel, and large enough so that the minimum radius of curvature is of the order of one channel width. The ideal value would be 2*pi, and restricting it simply anticipates the expectation that paths containing many loops will ultimately be rejected. In other words, this keeps paths close to the locally optimal direction, but allows some latitude so that branches can form.

   The angle between new paths (pi/24) is another finite difference parameter – we cannot consider the continuum of angles spanning (-pi/6,pi/6). Again the ideal value would be zero and we reduce it progressively until the same (or similar enough) set of paths are generated.

   The values |x-xi|, |y-yi| < 16 km are chosen to identify similar paths. The first two are chosen such that one of a set of seeds is identified within a small number of generations, if the paths that start from from them appear to be converging.

   The condition |a-ak| < pi/8 is also chosen to identify similar paths. The angle must be greater than zero to allow branches to form from paths which are identical up to the point of the branch, and persist provided they arrive at distinct end point. A smaller angle reduces the number of branches in play at any one generation.

   We have added additional text to section 3.1 to add further clarity.

10. *page 8, l. 7: I think the explanation for step 7 is easier to understand if the words "When considering the length of all complete paths," are removed and the sentence simply starts with "Where …".*

    Done.

11. *page 9, l. 17: I think there should be no comma after "parameters".*

    Removed.

12. *page 13, l. 22: I suggest to remove the paragraph break here. Maybe the sentence in l. 20-22 can be rewritten to clarify that the improvement is an improvement compared to Bed2013, with OBS1516 as a reference what the targeted truth is.*

    The paragraph break has been removed and this has now been rewritten:

    "Considering centreline profiles for all fjords, we illustrate the improvements made to the general elevation profile of each fjord (Fig. 12) relative to those present in Bed2013, by considering the general agreement between the synthetic geometry and OBS1516."

13. *page 14., l.1: I think "provided" or "presented" is better here than "illustrated"*

    We now use presented.

14. *page 15, l. 10: "_at_ a significant distance" ?*

    We now write "…at a significant distance…"

15. *page 16, l. 21: "high-frequency" ?*

    We now ensure all uses of "high frequency" are hyphen-free.

16. *page 16, l. 34: should it be "_for_ Bed2013" ?*

    Yes – this is now corrected.

17. *caption to Figure 5: Even with the electronic version in front of me, the "Please refer to the online version of this article to make use of references to colour." bit fails to make sense to me.*

    This has been removed.

18. *caption to Figure 8: I suggest to make it "along-transect" at both locations*

    Done.

19. *Figure 9: The figure shows that SynthIBCAO and SynthOBS give very similar results, which is good and demonstrates the ability of the method. However, I miss a possibility to directly compare to the OBS1516 data. Would it be possible to replace the grey area in panel a, which just indicates data coverage, by the OBS1516 elevation data? Keeping the extent / data coverage information of course (white areas remain white).*

    Yes – this is now implemented.

*20. caption to Figure 11: I am sure the last sentence can be formulated in an easier way (still making the point to be made)*

We have reworded this to:

"Positive differences (red) occur where the subtrahend is deeper than the minuend, with negative differences (blue) occurring where the subtrahend is shallower than the minuend."

*21. In general, I miss a statement (e.g. in the Summary) on how many manual steps are required during the procedure. Manual steps are mentioned at several locations in the manuscript, but I wonder how much work it would be to do this for, let's say, entire Greenland.*

We have added this to the end of the discussion:

"This paper provides a proof of concept routine for constructing geomorphologically realistic fjord geometry in the absence of observations. Actual implementation of the presented routine for large regions (e.g. the Greenlandic coast) would require manual intervention in so far as (i) identifying a seed elevation at the head of the channel and (ii) defining an end zone (e.g. the fjord mouth). Step (i) could be achieved by using a nearest neighbour approach to acquire the nearest elevation to a given seed location. A solution to step (ii) could be by using an observation density grid where the end zone is identified as being a location with an observation density greater than a chosen value. In addition to this, the values necessary to prevent the development of closed circuit artefacts would have to be adapted to the width of the fjords for which the method is implemented."

In addition to the changes made in response to the above comments, we have made some additional amendments.

Fig. 10: We have adjusted the panels and the coverage in (a) is now slightly more restricted than was presented in the original manuscript – the new coverage does not affect any of the analyses within the presented manuscript other than the reported statistics with regard to differences between Bed2013 and OBS1516 (section 4.1). We have also corrected the description of the under- and over-estimates indicated by the colour scale bar.

Fig. 13: Improved resolution

---

## Author Response (AR2)

Dear Oliver,

Many thanks for your comments. We have made the additional corrections, highlighted in the uploaded marked up manuscript - details of the changes are as follows:

- page 5, line 27 (on the version attached to the reply to reviews): don't understand the 'which cannot' in the brackets.

This was incomplete and the following now replaces it:

(i.e. the white regime, which cannot be stochastically modelled)

- page 8, line 6 (and elsewhere): table 1 -> Table 1

Altered all instances from table -> Table

- page 8, line 12: Fig. 5(b) should be Fig. 5a (and at 2 other places in this page)

Altered on four occasions on page 8, and also, we now make reference to the new Fig. 5(b):

"see Fig. 5(a) for a straight fjord example and Fig. 5(b) for a curved fjord"

- page 9, lines 4-7: the text seems in contradiction with Fig. 6 on which I can only see normal calculation using the two first neighbours? There is no illustration showing a calculation using more distant neighbours, as seems to be suggested by this sentence?

Fig. 6 does not illustrate the case of more distant neighbours and the reference to the figure in relation to this profile smoothing has now been removed.

- page 11, Results: check that the numbering of figures follows their order of appearance. Seems that it is not respected here.

The text has been altered and the positions of the original Fig. 9 and 10 have been switched to ensure the figure order now follows the order of appearance. We have also added text to the caption of Fig. 8 to improve clarity with regard to its link with Fig. 7.

Figures 10 and 11 are introduced as 10a/10b/10c + 11a/11b followed by 10d + 11c/11d. This is because both figures 10 and 11 show equivalent difference routines for 2 regions, the regions being discussed in consecutive sections (4.2 + 4.3). Figures 10 and 11 are grouped so to keep the specific differencing approaches together.

**Other changes**

All case of overestimation and underestimation changed to -> over-estimation and under-estimation

pg 3, In 20: modeling -> modelling

pg 3, In 25: modeled -> modelled

pg 5, ln 1: derived from combination -> derived from a combination

pg 5, ln 10: U-shape -> u-shape

pg 6, In 14: (Rignot et al., 2016, OMG Mission 2016) the data being -> (Rignot et al., 2016, OMG Mission 2016), the data being

pg 6, In 15: based on two datasets - IBCAO -> based on two datasets -- IBCAO

pg 6, ln 24: presence of a centreline - an imaginary -> presence of a centreline -- an imaginary

pg 8, ln 22: paths 1.11.1, 1.11.2 and 1.11.3 complete, the shortest where L is minimised (1.11.2) being retained -> paths 1.8.1, 1.8.2 and 1.8.3 complete, the shortest where L is minimised (1.8.2) being retained

pg 9, ln 19: (taken from OBS1516 (Fig 3)) -> (taken from OBS1516 (Fig. 3(b)))

pg 9, ln 25: (-920 m -- see Fig. 3) -> (-920 m -- see Fig. 3(b))

pg 10, ln 27: generate synthetic profile -> generate a synthetic profile

pg 14, ln 24: illustrated in Fig. 7 - > illustrated earlier in Fig. 7(b)

pg 15, In 10-11: power spectra in Fig. 14, and the stochastic inverse Fourier transform procedure describe -> power spectra in Fig. 13, and the stochastic inverse Fourier transform procedure described

pg 16, In 3: albeit that -> however

pg 16, In 15: these -> those

pg 17, ln 12: statistical features of these features - > statistics of these features

pg 18, In 20: "and CNW" added

Fig. 3(b) - reference removed - data here is specifically from the OMG Mission

Fig. 9(c) – x axis tick label overlap removed

Kind regards,

**Chris Williams**

**Generating synthetic fjord bathymetry for coastal Greenland**

Christopher N. Williams1, Stephen L. Cornford1, Thomas M. Jordan1, Julian A. Dowdeswell2, Martin J. Siegert3, Christopher D. Clark4, Darrel A. Swift4, Andrew Sole4, Ian Fenty5, and Jonathan L. Bamber1 1Bristol 
[revised manuscript text omitted]